# Reactivatable stimulated emission depletion microscopy using fluorescence-recoverable nanographene

Qiqi Yang [1], Antonio Virgilio Failla[2], Petri Turunen[3], Ana Mateos-Maroto[1], Meiyu Gai[1], Werner Zuschratter[4], Sophia Westendorf[5], Márton Golléri [3], Qiang Chen [1], Goudappagouda[6], Hao Zhao[6], Xingfu Zhu [1], Svenja Morsbach [1], Marcus Scheele [5], Wei Yan[7], Katharina Landfester [1], Ryota Kabe [8] ✉, Mischa Bonn [1] ✉, Akimitsu Narita [1,6] ✉ & Xiaomin Liu [1] ✉

Stimulated emission depletion (STED) microscopy, a key optical super-resolution imaging method, has extended our ability to view details to resolution levels of tens of nanometers. Its resolution depends on fluorophore de-excitation efficiency, and increases with depletion laser power. However, high-power irradiation permanently turns off the fluorescence due to photobleaching of the fluorophores. As a result, there is a trade-off between spatial resolution and imaging time. Here, we overcome this limitation by introducing reactivatable STED (ReSTED) based on the photophysical properties of the nanographene dibenzo[*hi,st*]ovalene (DBOV). In contrast to the photo-induced decomposition of other fluorophores, the fluorescence of DBOV is only temporarily deactivated and can be reactivated by near-infrared light (including the 775 nm depletion beam). As a result, this fluorophore allows for hours-long, high-resolution 3D STED imaging, greatly expanding the applications of STED microscopy.

Stimulated emission depletion microscopy (STED)[1,2] allows imaging with nanometer resolution[3]. It narrows the focal point spread function (PSF) by superimposing an additional red-shifted structured light beam, e.g., commonly used doughnut-shaped depletion beam on a Gaussian-shaped excitation beam to minimize the detectable spontaneous emission area[4]. The involved fluorophores should thus fulfill the following criteria: (i) the stimulated emission by the STED beam must lead to efficient depletion of the excited fluorescent state; (ii) the fluorophore must exhibit a high photo-stability/resistance to bleaching against STED and excitation beams; and (iii) the absorption and emission spectra should have minimal overlap to avoid adverse re-excitation by the STED beam[5,6]. There is an intrinsic trade-off in STED microscopy: a high-power STED beam is needed to achieve high resolution, yet high laser powers commonly result in photo-bleaching, most problematically the photo-induced decomposition of the fluorophores[7]. Such photochemical alteration of the fluorophore molecules includes a permanent loss of fluorescence and severely affects the imaging quality, especially the ability to perform long-term or 3D imaging. Protective agents, e.g., commercially available anti-fading agents, are therefore customarily needed as a stabilizer to

[1]Max Planck Institute for Polymer Research, Mainz, Germany. [2]UKE Microscopy Imaging Facility, University Medical Center Hamburg-Eppendorf, Hamburg, Germany. [3]Institute of Molecular Biology gGmbH, Mainz, Germany. [4]Leibniz Institute for Neurobiology, Magdeburg, Germany. [5]Institute of Physical and Theoretical Chemistry, University of Tuebingen, Tuebingen, Germany. [6]Organic and Carbon Nanomaterials Unit, Okinawa Institute of Science and Technology Graduate University, Onna-son, Okinawa, Japan. [7]Key Laboratory of 3D Micro/Nano Fabrication and Characterization of Zhejiang Province, School of Engineering, Westlake University, Hangzhou, Zhejiang Province, China. [8]Organic Optoelectronics Unit, Okinawa Institute of Science and Technology Graduate University, Onna-son, Okinawa, Japan. ✉e-mail: ryota.kabe@oist.jp; bonn@mpip-mainz.mpg.de; akimitsu.narita@oist.jp; Liuxiaomin@mpip-mainz.mpg.de

inhibit rapid photo-bleaching of the fluorophores[8–10], but can limit the applicable imaging environment[11]. Other approaches aim to enhance the resolution at lower depletion power levels[12,13] or develop fluorescent probes for high-power STED microscopy[14–18].

In this work, we report that dibenzo[*hi,st*]ovalene with two mesityl (Mes) groups (DBOV-Mes), as a red-emissive nanographene (NG), can overcome the photo-bleaching issue of STED imaging. The high-power STED beam does not photo-bleach the fluorescence of DBOV-Mes, but instead recovers the fluorescence that is gradually deactivated by the excitation laser. This property leads to reactivatable STED (ReSTED) imaging technology. Moreover, combined with its other properties, such as high brightness[19], superior chemical- and photo-stabilities, and environment-independent fluorescence behaviors, DBOV-Mes is particularly useful for long-term 3D STED imaging without the need for an anti-fading medium.

## Results

### Photophysical properties of DBOV-Mes

Nanographenes (NGs), molecular graphene nanomaterials with atomically precise and designable chemical structures, have emerged as promising candidates for imaging probes[20–22], owing to their well-known fluorescence properties and high photo-stability[23,24]. In the NG family, DBOV and some other structures with zigzag edges have demonstrated stimulated emission[19,25,26], and have been used for lasing applications[27,28]. Their stimulated emission properties should also benefit STED imaging, although it has not yet been investigated. In particular, the 775 nm STED beam typically used in commercial STED microscopy setups is located on the red side of the DBOV-Mes emission spectral range and does not coincide with the absorption range (Fig. 1a). Therefore, we choose DBOV-Mes as an example to demonstrate NGs for STED imaging and the 561 nm beam is applied for excitation based on the absorption spectrum of DBOV-Mes (Fig. 1a).

First, we demonstrated that the high-intensity STED beam has negligible photo-bleaching/quenching effect on DBOV-Mes molecules −before and after continuous irradiation with the STED beam, the measured fluorescence intensity varied little (Supplementary Fig. 1). This might be attributed−similar to other NGs−to the high chemical stability of DBOV-Mes, originating from the rigid $\pi$-conjugated structures with considerable aromatic stabilization[29].

Second, we tested the photo-stability of DBOV-Mes under excitation-beam-only (e.g., 561 nm) conditions. As shown in previous studies[21,30,31], even though the photo-stability of NGs is superior to that of organic fluorophores, their fluorescence intensity still decreases gradually due to the excitation beam. The same behavior applies to DBOV-Mes, as shown in both Supplementary Fig. 2 and Fig. 1b−d. After 100 consecutive frames of confocal imaging of a crack area of the coverslip after dropping the DBOV-Mes (see Methods for sample preparation), nearly 50% of the fluorescence intensity was quenched/deactivated (region of interest (ROI)1 of Fig. 1b, c). Here, we refer to the behavior of the fluorescent molecules gradually entering a fluorescence off state when exposed to an excitation beam as "deactivation".

We found that the deactivated fluorescence of DBOV-Mes can be recovered by the STED/depletion beam. Here, the depletion beam (775 nm) in the STED microscope is used to recover fluorescence, hereafter referred to as the reactivation beam. For example, one high-power reactivation beam was used to scan ROI2−the orange dashed box area in Fig. 1c after the deactivation (Supplementary Note 1). The confocal image taken subsequently (Fig. 1d) shows that all fluorescence was recovered. Representative line profiles clearly show the intensity change (Supplementary Fig. 3). We refer to the behavior of the non-fluorescent molecules returning to the fluorescence on state when exposed to a reactivation beam (e.g., 775 nm) as "reactivation".

We propose photoinduced ionization and photo-stimulated recombination as the deactivation and reactivation processes (Fig. 1e). Upon excitation by the 561 nm laser, DBOV-Mes molecules are rapidly excited from the singlet ground state ($S_0$) to the singlet excited state ($S_1$), and exhibit fluorescence with nanoseconds lifetime from the $S_1$. Under continuous irradiation, the molecule repeats several excitation/recombination cycles between the $S_0$ and $S_1$ states, which defines the fluorescence on state. On the other hand, the molecule can release one electron by photoionization to form the radical cation of DBOV-Mes (DBOV-Mes$^{\cdot+}$). Radicals are generally non-luminescent states. They are stable unless they react with different materials or return to the neutral state by recombining with electrons[32], so DBOV-Mes$^{\cdot+}$ functions as the fluorescence off state. Although direct photoionization does not occur with a 561 nm laser as the ionization potential calculated from cyclic voltammetry is around 4.49 eV[33], ionization by resonance-enhanced multiphoton ionization (REMPI)[34] with $S_1$ or triplet excited state ($T_1$) as intermediate states (or non-resonant two-photon ionization) is possible if the laser intensity is sufficient (Supplementary Fig. 4)[35,36]. The recombination process of radical cations and electrons occurs over time but can be accelerated by photostimulation[37]. This is because DBOV-Mes$^{\cdot+}$ (or the radical anion of the medium that received the released electron) has a characteristic absorption of radicals in the NIR region (Fig. 1a, Supplementary Figs. 5 and 6 for details). As a result, the photostimulation by the reactivation beam causes charge recombination and the DBOV-Mes are again in the fluorescence on state.

We note here that during STED imaging, high depletion laser intensities ($10^{1–3}$ MW cm$^{-2}$)[38,39] are required to obtain high resolution. Such high laser intensities produce a much larger optical density distribution in the focal imaging plane (Supplementary Fig. 7). Thus, even if the reactivation beam scans only a small region (ROI2 of Fig. 1c), molecules in a larger area (Fig. 1c ROI 1 over ROI2) can also be reactivated by receiving non-negligible amounts of photons. Besides, in many of our measurements, the fluorescence intensity after reactivation by the reactivation beam (e.g., ROI2 of Fig. 1d) was even higher than in the first image (before deactivation) (e.g., Fig. 1b). This is due to factors such as ambient light which is difficult to avoid and the beam excitation that is required to find the focal plane and select the target imaging area before imaging. Therefore, before taking the first confocal image (e.g., Fig. 1b), some DBOV-Mes molecules may already be in the fluorescent off state.

### Reactivation properties of DBOV-Mes

To corroborate the deactivation and reactivation processes proposed in Fig. 1e and to quantify such properties of DBOV-Mes, we performed a series of experiments. Figure 2a shows two imaging modes used for the different experiments. First, in studying the deactivation process, only the excitation beam of 561 nm was used, and confocal images were acquired continuously (Fig. 2a(1)). One can see that the fluorescence intensity $F(t)$ (insets of Fig. 2a(1)−confocal images of different imaging frames/times) decays exponentially over time. Further, to study the relationship between the decay rate and the excitation laser intensity, we measured the fluorescence decay/deactivation curves under different excitation laser intensities at different times (Supplementary Fig. 8), and analyzed the dependence of the fluorescence ratio on the excitation power density. A linear plotted curve with a slope of 2 was obtained in the region of shorter excitation times (total lower excitation intensity) (Fig. 2b, after 30 frames of excitation), supporting the two-photon ionization process (see Supplementary Note 1 for experimental details; Methods and Supplementary Fig. 9 for data analysis details). Note that due to the saturated excitation, the slope gradually decreases and approaches 1 after 150 frames.

Next, in studying the reactivation process, as shown in Fig. 2a(2), three confocal images were recorded−before ($D_1$) and after ($D_2$) deactivation by the excitation beam of 561 nm, and after reactivation ($D_3$) by the reactivation beam of 775 nm. The dependence of the fluorescence reactivation rate on the reactivation beam intensity was investigated and a slope of ≈1 was fitted from the lnln-ln plot of

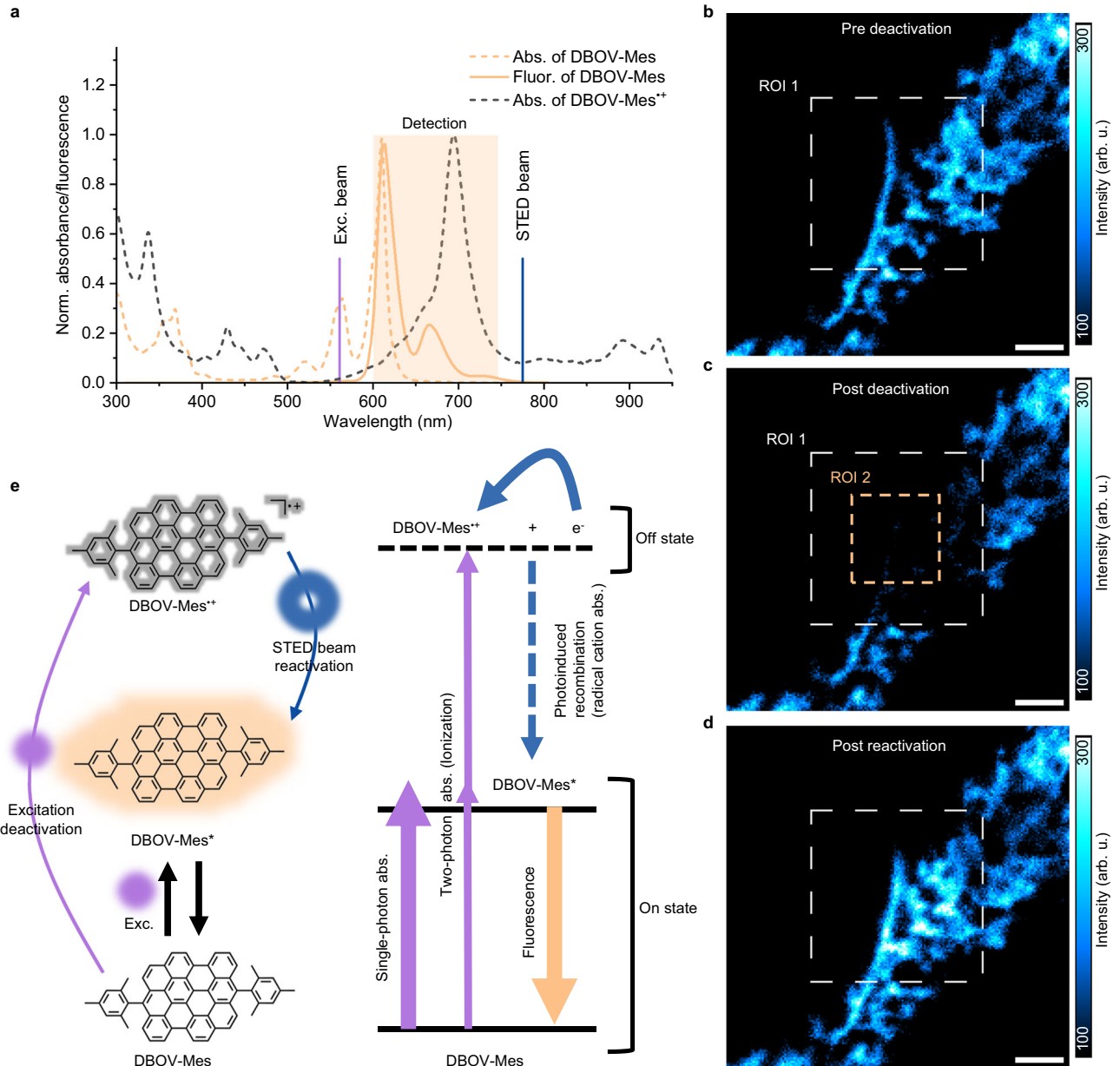

**Fig. 1 | ReSTED using fluorescence-recoverable nanographene. a** Normalized absorption and emission spectra of DBOV-Mes (µM in toluene) and absorption spectrum of DBOV-Mes⁺ after the addition of 10 equivalents of SbCl₅ to a solution of DBOV-Mes in dry dichloromethane. Excitation (Exc.) beam: 561 nm; STED beam: 775 nm; detection range: 600–750 nm. Confocal images of cracks from a coverslip with DBOV-Mes before (**b**) and after (**c**) deactivation by excitation beam and after reactivation by the reactivation beam (**d**). White dashed square region – region of

interest (ROI)1: deactivated area. Orange dashed square region–ROI2: reactivation beam scanned area. Scale bars: 2 µm. The color bars represents a linear scale. **e** Diagrams illustrating molecular states (ground state DBOV-Mes, fluorescent excited state DBOV-Mes*, and non-fluorescent deactivated state DBOV-Mes⁺) of DBOV-Mes for ReSTED imaging. All the parameters are summarized in Supplementary Note 1. Source data are provided as a Source Data file.

unrecovered fluorescence ratio versus reactivation power density (as shown in Fig. 2c), supporting one photon reactivation process (see Supplementary Note 1 for experimental details and Methods and Fig. 9 for data analysis details).

Furthermore, we characterized the repeatability of such deactivation and reactivation processes. After 7 cycles of repeated short-term deactivation and reactivation (similar to the STED imaging setup), no significant reduction in fluorescence was observed. In contrast, for comparison, in the absence of reactivation, there was a 30% reduction in fluorescence (Fig. 2d). On the other hand, as shown in Fig. 2e, such repeated reactivations can also be achieved during longer periods of deactivation and activation, but with a superimposed overall exponential decay (see Supplementary Note 1 for experimental details and

Methods for data analysis details). This attenuation may be caused by the multiphoton effect resulting from the high peak laser intensity of the pulsed reactivation beam. For example, two-photon fluorescence was observed when increasing the reactivation beam peak intensity (Supplementary Figs. 10, 11), which might compete with the single-molecule reactivation process. The same trend was also observed at different deactivation rates (Supplementary Fig. 12). This may also be the reason for the fluorescence reduction observed after long-term excitation with a high-power pulsed reactivation beam in another experiment (Supplementary Fig. 13). To address these issues, one solution could be the use of longer pulse duration or even continuous wave (CW) reactivation beam, which requires further investigation in the future and is beyond the scope of the current work.

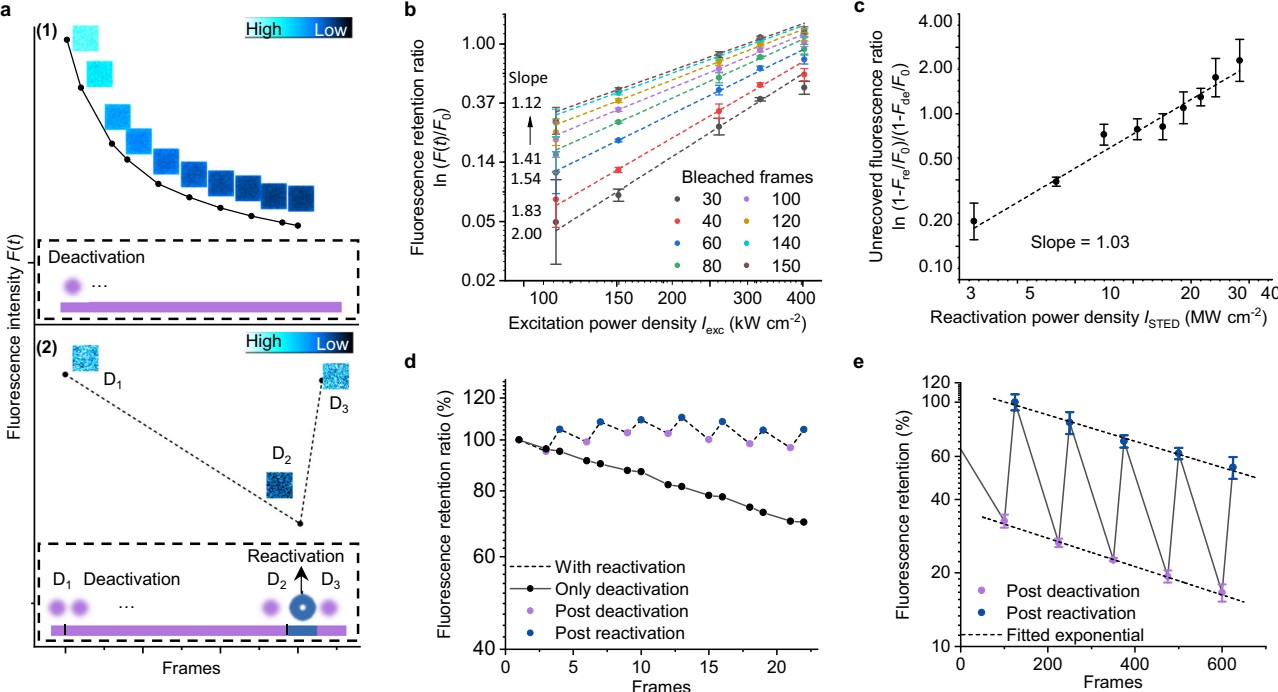

**Fig. 2 | Reactivation properties of DBOV-Mes. a** Two modes were used to characterize the reactivation properties. The diagram in dashed boxes shows the sequences of beams and fluorescence detection. **1** For deactivation, only the excitation beam was used, and confocal images were acquired continuously. Insets: confocal images of different imaging frames/times. **2** For reactivation, three confocal images were recorded, before ($D_1$) and after ($D_2$) deactivation by excitation beam and after reactivation ($D_3$) by reactivation beam. The size of all the images in **a** is 6.95 μm × 6.95 μm. The color bar represents a linear scale. **b** Plot (lnln–ln) of leftover fluorescence ratio ($F(t)/F_0$) against excitation power density ($I_{exc}$) over different imaging frames/time. Plot (lnln–ln): Plot of the double logarithmic transformation of the leftover fluorescence ratio ($F(t)/F_0$) against the natural logarithm of the excitation power density. The equations can be found in "Methods" part. The solid circles of different colors are the measured data, and the dash lines are the fitted linear curves with different slopes. $F(t)$: measured emission intensity at the frame time $t$; $F_0$: measured emission intensity at the initial frame time. **c** Plot (lnln–ln) of reactivation fluorescence ratio against reactivation beam power density. Plot (lnln–ln): Plot of the double logarithmic transformation of the reactivation fluorescence ratio against against the natural logarithm of the reactivation beam power density. The equations can be found in Methods part. The dashed line represents the fitted linear curve with the slope ≈1. **d** Dashed line: measured fluorescence after repeated short-term deactivation and reactivation cycles. Green dots: fluorescence after deactivation; Red dots: fluorescence after reactivation. Solid line (black dots): fluorescence with only deactivation. **e** Fluorescence retention after repeated long-term deactivation and reactivation cycles. Dashed lines: fitted exponential fluorescence decay curves. Error bars in (**a**, **c**, **e**) represent standard deviations calculated from 3 or 4 measurements. All the experimental settings/parameters are summarized in Supplementary Note 1. Source data are provided as a Source Data file.

## DBOV-Mes shows superior properties for STED imaging

As a next step, we performed more experiments to characterize and study the properties of DBOV-Mes for STED imaging. First, to investigate the reactivation efficiency, imaging without (confocal imaging) and with reactivation (STED imaging−combined excitation beam and STED/depletion beam) were recorded. As a control, we compared the ratio of leftover fluorescence before and after imaging (Supplementary Fig. 14). As shown in Fig. 3a, b, large confocal images were recorded before (Fig. 3a) and after (Fig. 3b) confocal/STED imaging. For the imaging, two smaller ROIs of the same size were imaged separately with a series of confocal or STED images. As expected, the leftover fluorescence intensity after STED imaging (right ROI) is higher than the one after confocal imaging (left ROI), due to the reactivation effect of the STED beam.

By increasing the number of imaging frames (total imaging time), we can characterize the reactivation efficiency by comparing the leftover fluorescence intensity after imaging with and without reactivation. One can see that there is always more fluorescence retained after STED imaging than confocal imaging (Fig. 3c). For example, after 300 frames, 94% of the molecules are quenched/deactivated after confocal imaging, while 47% of the fluorescence remained after STED imaging. It should be noted that while STED light can enable fluorescence recovery of DBOV-Mes, very high STED power may cause bleaching due to nonlinear effects, making it essential to balance STED power for optimal fluorescence recovery and minimal bleaching.

Second, reactivation beam power-dependent fluorescence reactivation experiments were performed to optimize STED imaging conditions (the best combination of excitation and reactivation beam powers), i.e., changing the reactivation beam power while keeping the excitation intensity constant. The results showed that after 150 consecutive STED imaging of the same imaging area, many of the DBOV-Mes molecules are still in the fluorescence ON state (Fig. 3d) under different STED power conditions. The higher the reactivation beam power, the higher the leftover fluorescence intensity. When the reactivation beam intensity approaches 38 MW cm⁻², the leftover fluorescence intensity decreases again, possibly due to the involvement of additional nonlinear processes (e.g., two-photon fluorescence excited by high power reactivation beam as discussed for Fig. 2e). We note here, all the above experiments were performed in an air environment.

Third, we further investigated the reactivation properties of DBOV-Mes under different imaging conditions (air, water, or PBS buffer) (Supplementary Fig. 15). As shown in Fig. 3e, in air and PBS conditions, all the fluorescence can be recovered by the reactivation beam. While in water, it is difficult to achieve 100% reactivation, which may be because after the ionization of DBOV-Mes, the electrons react with water and cannot recombine.

Besides, in STED imaging, a Gaussian excitation beam and a doughnut-shaped depletion beam are used, and it is difficult to analyze the effect of photons at these two wavelengths on DBOV-Mes when both beams scan the same area. Here, we applied two temporally and

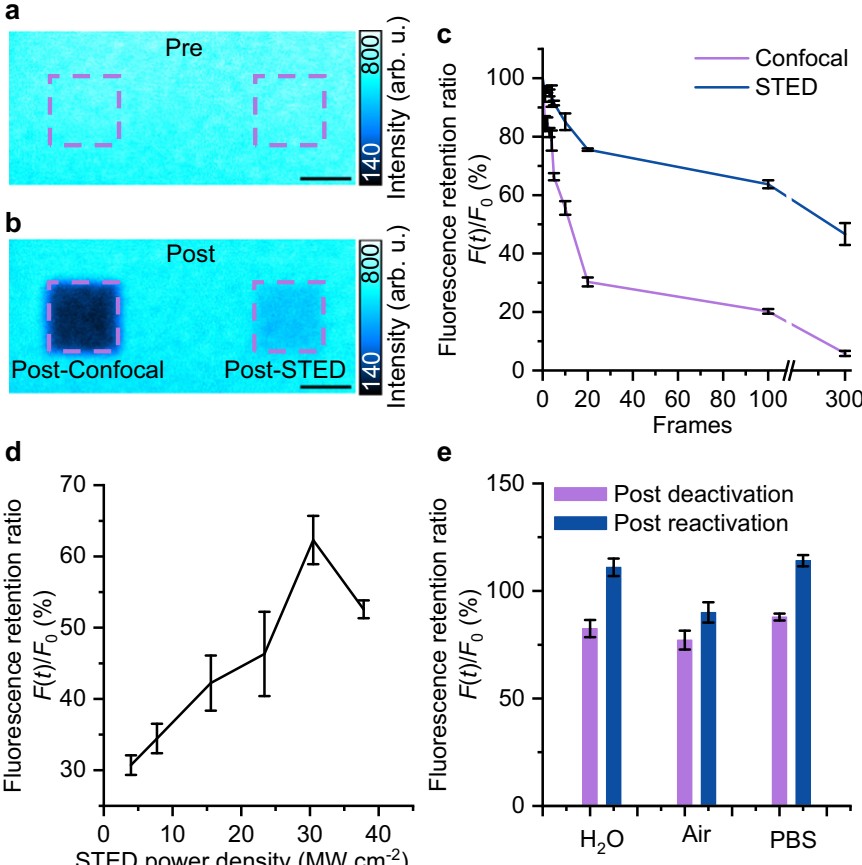

**Fig. 3 | Reactivation properties of DBOV-Mes. a, b** Confocal images of before (upper panel) and after (lower panel) confocal/STED imaging. The ROI in red dashed boxes was imaged by confocal and STED imaging for 5 frames. Scale bars: 5 μm. The color bars represent a linear scale. **c** The leftover fluorescence ratio after confocal and STED imaging over the number of imaging frames/time. **d** The leftover fluorescence intensity dependent on the reactivation beam power density while keeping the excitation intensity constant. **e** The reactivation properties of DBOV-Mes measured under different imaging conditions (air, water, or PBS buffer). Error bars in b-d represent standard deviations calculated from 3 or 4 measurements. All the experimental settings/parameters are summarized in Supplementary Note 1. Source data are provided as a Source Data file.

spatially overlapped Gaussian pulse beams (of excitation 561 nm and depletion 775 nm wavelengths), illuminating the DBOV-Mes sample, and the results show negligible photo-bleaching effect (Supplementary Fig. 16).

Moreover, the absorption spectrum of the OFF state (radical cation) of DBOV-Mes was measured and verified by two methods - testing the absorption spectrum after chemical oxidation in solution and recording the change in absorption during electrochemical oxidation via potential-modulated absorption spectroscopy (EMAS) (see Supplementary Information for details of these two methods). The two experimental results are consistent with each other. In brief, after oxidizing by an excess amount of oxidant $SbCl_5$, new absorption peaks appear as shown in Supplementary Fig. 5. On the other hand, in EMAS experiments, we find that applying an oxidative potential of $+(0.15 \pm 0.1)$ V vs. the ferrocene/ferrocenium redox couple is sufficient to bleach the characteristic absorption bands of DBOV-Mes at 573 and 618 nm, which we assign to the chemical transformation into the DBOV-Mes$^{\cdot+}$ radical. At the same potential, we observe broad induced absorption bands centered at 440, 708 and above 800 nm (Supplementary Fig. 6). We note that the latter bands are nearly resonant with the reactivation beam (775 nm), such that their excitation provides a probable pathway for a photo-induced reduction to the neutral DBOV-Mes species in line with the scheme in Fig. 1a.

Based on the absorption spectrum of OFF state of DBOV-Mes, we tested two other laser beams with wavelengths of 730 and 405 nm

(lasers available in our microscope setups), which are also within the absorption spectral range of DBOV-Mes$^{\cdot+}$, for reactivation experiments. The results show that both lasers can be used to recover the deactivated fluorescence (Supplementary Figs. 17, 18). We note here that the reactivation is not sensitive to the concentration of DBOV-Mes molecules (Supplementary Fig. 17). In addition to photo-induced fluorescence reactivation, another experiment suggests that the fluorescence recovery process may include a slow temperature-dependent thermodynamic process. After deactivation, a part of the radical cation DBOV-Mes$^{\cdot+}$ can spontaneously return from the OFF state to the ground state under room temperature and dark conditions – around 30% of the fluorescence signals were recovered after 14 h (Supplementary Fig. 18). This suggests that fluorescence could be spontaneously recovered, which is however much less efficient than light-induced electron recombination.

## ReSTED imaging
As proof of concept of the feasibility and superiority of ReSTED with DBOV-Mes, we performed 2D and 3D imaging of suitable model systems. For 2D imaging, DBOV-Mes was embedded in polystyrene (PS) film to form nanoparticles. As shown in Fig. 4a–d, the STED images clearly show an improvement in resolution, which enables the visualization of delicate structures that emerge from more intertwined regions as well as other subtle features. The line profiles in Fig. 4e correspond to the dashed lines in the right panels in Fig. 4b, d. They demonstrate the improved resolution of the STED image, with full

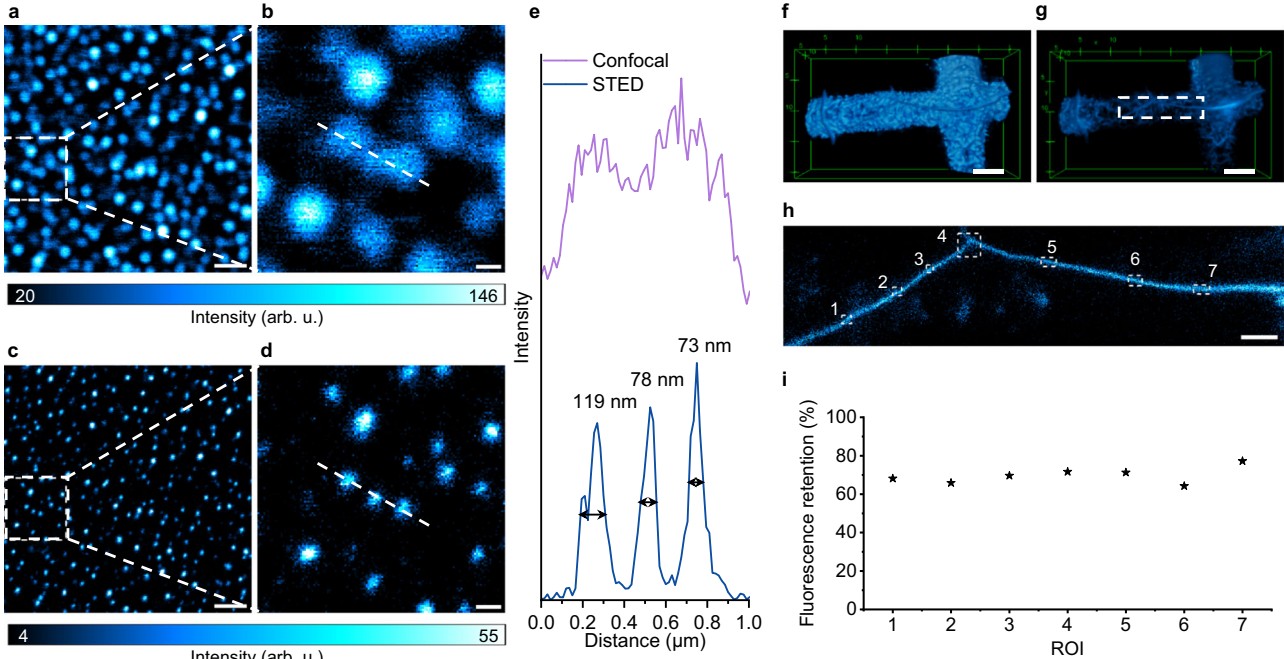

**Fig. 4 | STED imaging of DBOV-Mes.** Confocal (**a**, **b**) and STED (**c**, **d**) images of DBOV-Mes embedded in PS film. The regions enclosed by the dashed boxes in (**a**, **c**) are magnified in the corresponding right (**b**, **d**). Scale bars: **a**, **c**, 1 μm; **b**, **d**, 200 nm. The color bars represent a linear scale. **e** Normalized intensity profiles along the dashed lines in (**b**, **d**). 2D-3D (2D STED beam with *z*-stack measurements) confocal (**f**) and STED (**g**) images of nanoscale cracks from the gridded structures in a glass substrate with DBOV-Mes. Scales, 5 μm. **h** STED image of the white dashed square region of (**g**) at an imaging depth of 4.68 μm relative to the surface of the glass substrate. Scale bars: 1 μm. **i** Fluorescence retention of DBOV-Mes after two sequential 2D-3D measurements obtained from ROIs in (**h**). All the parameters are summarized in Supplementary Note 1. After repeated 3D imaging over the same imaging volume for more than 1 h, more than 70% of its STED fluorescence intensity was retained. Source data are provided as a Source Data file.

width at half maximum (FWHM) of 119, 78, and 73 nm from three separate peaks compared to the confocal image.

For long-term imaging, *z*-stack imaging experiments were carried out, in which nanoscale cracks of gridded coverslips were imaged in ambient air after deposition of DBOV-Mes. The sample was imaged by both confocal and 2D-STED at each *z*-position. Along the z-direction, a series of 2D images on the *xy* plane were acquired with a step size of 0.15 μm. Finally, 3D volumes were reconstructed with the image processing software ImageJ. Figure 4f, g shows the reconstructed 3D confocal and STED images. Both 2D and 2D-3D STED images show improved resolution compared to confocal images (Supplementary Movies 1, 2). Long-term (>1 h) continuous confocal and STED imaging in turns over the same imaging volume have been performed. DBOV-Mes demonstrated high photo-stability during the whole process. Compared with one former measurement (Supplementary Fig. 19), DBOV-Mes maintained ≈70% (Fig. 4h, i) of its STED fluorescence intensity. The total imaging time for the entire 3D volume of Fig. 4g was over 73 min. The high stability of DBOV-Mes measured in air without additional anti-fading medium, therefore, allows its use in imaging applications with few constraints on its environment.

Last but not least, ReSTED imaging of DBOV-Mes labeled biomimetic liposomes was performed. Liposomes, self-assembled vesicles formed from amphiphilic (phosphate) lipids, are well developed as essential nanoscale drug delivery systems[40–42], which are currently regarded as one of the most clinically acceptable nano-formulations for disease treatment[43]. Unilamellar liposomes can be subdivided into three main categories based on size: The term small unilamellar vesicles (SUVs) is often used for vesicles of 20–100 nm diameter, large unilamellar vesicles (LUVs) for 100 nm –1 μm vesicles and giant unilamellar vesicles (GUVs) for vesicles of >1 μm diameter. The optimal size range of liposomes for drug delivery, typically 20–200 nm in diameter[44], is out of the diffraction limit of light microscopy.

Therefore, labeling with appropriate dyes for SRM imaging is crucial for the visualization of individual liposome vesicles and several application possibilities, e.g., tracking the internalization pathways of liposomes into cells and their final distribution. Currently, resolution limitations, for example, render it very difficult to distinguish between liposome-cell membrane fusion and endocytosis[45]. Moreover, the morphology of liposomes and the morphological dynamics in response to various environments such as temperature, osmotic pressure, pH and surfactants are essential for studying the ways of drug delivery and the visualization of liposome-cell interaction and drug distribution[46,47]. Figure 5a shows the representative deformation processes of liposomes (SUVs), resulting in deformed, and larger unilamellar vesicles or muti-lamellar vesicles.

In this work, the hydrophobic DBOV-Mes molecules are applied for liposome labeling and reside in the lipid tail domain of the bilayer membrane of SUVs (see preparation and basic characterization of the liposomes in Methods and Supplementary Figs. 20, 21). The prepared liposomes were finally immobilized on a coverslip surface and imaged after adding PBS solution without any anti-fading medium (Supplementary Fig. 22). Accordingly, we characterized freshly prepared DBOV-Mes-labeled liposomes and liposomes stored in PBS solution (4–8 °C) for 1 year. With freshly prepared liposomes, only single vesicles were imaged (Fig. 5b, c, e, f). Figure 5b, c, e, f shows the comparison between confocal and 2D/3D STED imaging modes, respectively. The STED images shown were processed by the Richardson-Lucy deconvolution method. The line profiles in Fig. 5d, g, corresponding to Fig. 5b, c, e, f clearly indicate the improved resolution of STED images, with FWHM of 76 nm (2D) and 184 nm (3D) from STED images compared to confocal images. Notably, while DBOV-Mes is theoretically capable of achieving high resolution under intense STED laser, it is essential to use an appropriate tunable emission filter for enhancing the signal-to-noise ratio (SNR). As for the liposomes stored over a year,

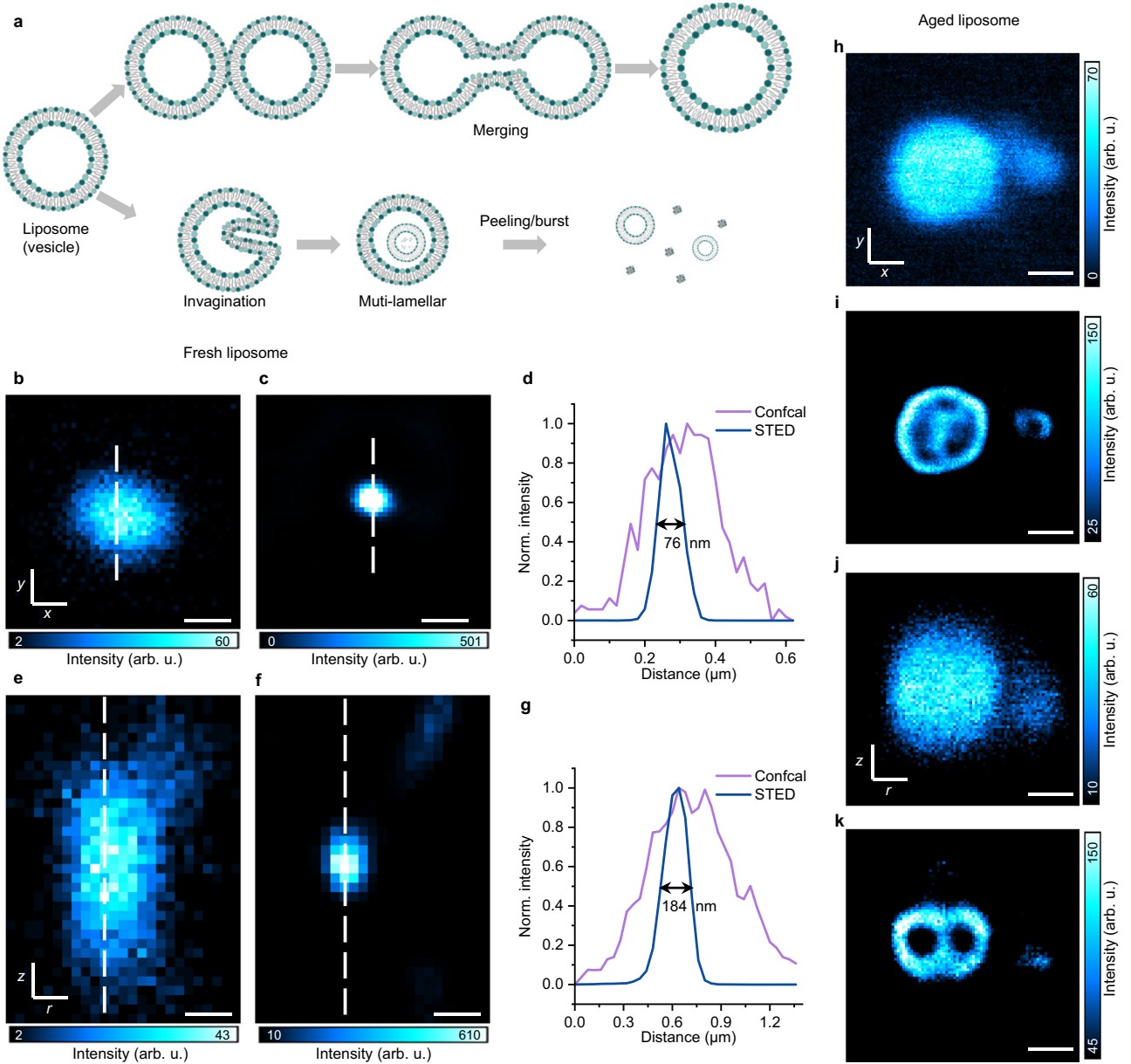

**Fig. 5 | 2D and 3D ReSTED imaging of DBOV-Mes-labeled freshly prepared and aged liposomes. a** Schematic diagram of representative deformation processes of liposome. 2D (**b, c**) and 3D-3D (3D-STED beam depletion in both the *xy* plane and along the *z*-axis) (**e, f**) Confocal (**b, e**) and STED (**c, f**) images of freshly prepared DBOV-Mes-labeled liposome. Scale bars: 200 nm. **d, g** Normalized intensity profiles along the dashed lines crossing a single liposome in (**b, c**) and (**e, f**). 2D (**h, i**) and 3D (**j, k**) Confocal (**h, j**) and STED (**i, k**) images of representative deformed DBOV-Mes labeled liposome (stored for 1 year in PBS solution). Scale bars: 500 nm. All the color bars represent a linear scale. All the parameters are summarized in Supplementary Note 1. Source data are provided as a Source Data file.

vesicles of different morphologies and sizes were formed. As shown in Fig. 5h–k, the enhanced STED resolution reveals the presence of clustered LUVs that are otherwise not distinguishable. The coalescence/merging and aggregation of LUVs can be clearly seen from the STED image in Supplementary Fig. 23. In addition, 2D and 3D STED images also confirm that small vesicles are enclosed within a large vesicle (GUV), forming vesicle-in-vesicle system called multivesicular vesicle (Supplementary Fig. 23). Notably, due to the high chemical and photo-stability of DBOV-Mes, labeled samples (e.g., liposomes) can be stored and imaged in their native conditions, which is rarely possible with other fluorophores.

## Discussion

Nanographene DBOV-Mes exhibits photophysical properties—its ON ground state can be recovered by red-shifted doughnut-shaped depletion beam excitation, thereby enabling ReSTED. The use of DBOV-Mes in ReSTED provides the opportunity for significant improvements in STED microscopy, thereby expanding the space of applicable parameters such as STED beam power and imaging time. The 3D ReSTED was carried out with both materials and biomimetic liposomes, using high STED power to achieve high resolution. The independence from an anti-fading medium will substantially expand the imaging applications of ReSTED, which will play an important role, e.g., in future live-cell and material imaging.

Meanwhile, we note that a variety of NGs have been synthesized and used in different applications—chiral NGs with near infrared emission for super-resolution imaging[48]; nitrogen doped DBOV for sensing and imaging (pH, heavy metal)[21]; water-soluble and bio-compatible DBOV for bio-imaging[49], etc. The spectra of DBOV derivatives and other types of nanographene obtained through organic

synthesis can cover the visible light region, and the narrow absorption and emission spectra make them ideal for multicolor imaging[50]. We believe that fluorescence-recoverable properties of DBOV studied in this work, combined with the above-mentioned advances, can further play an important role in related areas.

# Methods

## Materials

Nanographene (DBOV-Mes) was synthesized as described in refs. [50],[51]. All the other chemicals were used as received. 1,2-dio-leoyl- *sn*-glycero-3-phosphoethanolamine (DOPE) (purity ≥ 99%), L-α-phosphatidylcholine (egg PC, purity ≥ 99%) and cholesterol (purity ≥ 99%) were purchased from Merck, Germany. Phosphate buffered saline (DPBS-buffer, -Mg²⁺, -Ca²⁺) and chloroform (≥99% anhydrous) were procured from Merck, Germany, while ethanol (99.5%) was acquired from Carl Roth, Germany. Polystyrene (PS) ($M_w$ 280 kg mol⁻¹) was purchased from Merck, Germany. Toluene (anhydrous, purity ≥ 99%), tetrahydrofuran (anhydrous, purity ≥ 99%), 3-aminopropyltriethoxysilane (APTS, 99%) and di(*N*-succinimidyl)-glutarat (purity ≥ 97.0%) were purchased from Sigma-Aldrich. *N,N*-Dimethylformamide (DMF, purity ≥ 99.5%) was from Thermo Scientific.

## Coverslip cleaning

The coverslips were sonicated in 1% Micro 90 alkaline cleaning solution (International Products Corporation) for 15 min. Then the coverslips were rinsed three times with Milli-Q water, sonicated in ethanol for 10 min, and finally dried with nitrogen flow. Afterward, those coverslips were cleaned by an oxygen-plasma cleaner (250 W, 5–10 min).

## Preparation of nanographene on grid coverslip

We use a clean glass syringe to take 10 μL of DBOV-Mes solution (35 μM in toluene), deposit this on a clean grid coverslip (grid 50, #1.5H, 170 μm, ibidi GmbH), and wait for the solvent to evaporate before the measurement.

## Preparation of nanographene embedded in polystyrene (PS) film

A PS (0.08 mg mL⁻¹ in toluene) solution was mixed 1:1 (v/v) with the DBOV-Mes solution. 10 μL of DBOV-Mes-PS solution (35 μM) was spin-coated on the cleaned coverslip. The coverslip was first spun at 2000 rpm for 60 s. The sample was dried overnight before the measurement.

## Preparation of nanographene for measuring in H₂O and PBS

We use a clean glass syringe to take 10 μL of DBOV-Mes solution (35 μM in toluene), deposit this on a clean grid coverslip (grid 50, #1.5H, 170 μm, ibidi GmbH), and wait for the solvent to evaporate before the measurement. When do the imaging, 10 μL of H₂O/PBS was dropped on the sample.

## Liposome preparation[52]

For the preparation of the liposomes, the thin-film hydration method followed by extrusion was used. Thus, liposome samples were prepared using a mini extruder with a polycarbonate membrane (pore size of 800, 400, and 200 nm) and 1 mL syringes (Avanti Polar Lipids). First, the lipid films were prepared from mixtures of cholesterol (Chol), DOPE, and egg PC (stock solutions of 10 mg mL⁻¹ in chloroform, stored at −20 °C). To reach the liposome molar composition of eggPC:DOPE:Chol = 1:1:1, 208 μL eggPC, 192 μL DOPE, and 100 μL Chol were mixed. Additionally, 5 μL of 0.35 mM DBOV-Mes in toluene solution was added to the lipid mixtures. Then, the lipid mixtures together with 500 μL of chloroform with 1 vol.% EtOH were added in a 50 mL round-bottomed flask. Afterward, the dried lipid films were obtained by evaporating the solvent under low pressure with a rotary evaporator

(450 mbar and 3 mbar each for 30 min at 42 °C). Subsequently, liposomes were obtained by hydration of the lipid films by the addition of 1 mL of PBS buffer (0.1 M, pH = 7.4). Then the mixtures were stirred over night at 500 rpm and sonicated in a water bath for 20 min. Afterward, the liposomes were extruded through polycarbonate membranes with pore sizes of 800, 400, and 200 nm for 11 times at each pore size. Additionally, liposome samples were purified by centrifugation in triplicate at 20,000 × g, 4 °C for 30 min, redispersed in fresh PBS, and stored afterward at 4 °C until further use.

## Immobilization of liposomes on functionalized coverslip surface

A clean coverslip was immersed in a solution of APTS in dry THF (1 M) for 2 h. Afterward, the coverslip was rinsed in ethanol to remove the excess APTS. The coverslip was then dried with a strong nitrogen gas flow. We dissolved di(*N*-succinimidyl)-glutarat into DMF at a concentration of 1 mg mL⁻¹, also containing an equal molar concentration of triethylamine as base[53]. The solution was deposited onto the surface of the APTS-modified coverslip to completely cover it. It is allowed to react for 1 h at room temperature. After that, the surface was washed with DMF at least 3 times to remove excess di(*N*-succinimidyl)-glutarat. The coverslip was again dried with a strong nitrogen gas flow. The PBS solution containing liposomes ($8.45 \times 10^{11}$ particles mL⁻¹) was transferred onto the NHS-modified coverslip. And allowed to react for 4 h at room temperature. The unfixed liposomes were carefully washed away with PBS. The liposome-covered coverslip was then sealed with PBS on a glass slide for storage to be tested.

## Absorption spectra

UV-vis-NIR absorption spectra were recorded on a Perkin-Elmer Lambda 900 spectrometer at room temperature using a 10 mm quartz cell. All spectra were measured in anhydrous dichloromethane (DCM). A solution (2.5 mL) of DBOV-Mes was placed in a vial, then a specified amount of the $SbCl_5$ solution was added. A certain amount of extra anhydrous DCM solvent was added to make the total solution volume constant (3 mL) so that the concentration of DBOV-Mes stays the same in all measurements.

## Potential-modulated absorption spectroscopy (EMAS)[54]

The EMAS measurements were performed in a home-built spectro-electrochemical transmission cell, equipped with a silver wire pseudo-reference electrode, a coiled platinum wire counter electrode (both with a diameter of 1 mm) and an FTO-coated glass (Fluorine-doped Tin Oxide, Sigma Aldrich, 735167, 2.2 mm, 7 Ω in⁻²) as working electrode. The working electrode was drop-coated with 50 μL of a DBOV-Mes solution (20 mM in toluene) or with 300 μL of a ferrocene solution (330 mM in toluene) for reference measurements. The measurement cell was assembled, filled with 6 mL 0.1 M KCl/H₂O dist. and placed in the beam bath inside a faraday cage. The coated working electrode was then transilluminated by monochromatic light generated by a quartz tungsten halogen lamp (Apex 2 QTH-Lampe, Oriel Instruments) and selected by a monochromator (Cornerstone 130 Extended Range Monochromator; Oriel Instruments). Above 600 nm, a 570 nm long-pass filter (Newport 10CGA-570 Colored-Glass Alternative Filter) was introduced into the beam path to block components with a higher diffraction order. The transmitted light was detected by a Si-biased detector (DET36A, Thorlabs) and the signal passed to a lock-in amplifier (MFLI, Zurich Instruments). A bipotentiostat (CHI760E; CH Instruments) then applied a constant potential at the predetermined open circuit potential (OCP) of approximately −0.25 V versus Fc/Fc⁺ to the working electrode. In addition, a rectangular-shaped modulated potential with a modulation frequency of 57 Hz and a variable amplitude as an offset was produced by a signal generator (PSG9080, Joy-IT) and superimposed on the constant potential. The lock-in amplifier also received the modulated potential generated by the signal generator as a reference signal and searched for a modulated component in the

optical signal that oscillates at the same frequency. The phase relationship between the modulated potential and the modulated component in the optical signal determined whether the measured signal was an induced absorption or a bleach. During the EMAS measurement, the modulated potential applied to the DBOV-Mes sample on the working electrode was increased in increments of 20 mV and at each potential step a complete spectrum from 399 to 900 nm was recorded in steps of 3 nm. The iR-compensation by the integrated positive feedback function of the bipotentiostat was activated and the measurement sensitivity was set to 0.1 mA V$^{-1}$. The AC-coupled signal input of the lock-in amplifier was set to 10 MΩ with an input range of 10 mV. Incoming signals were processed by a 6th-order low-pass filter having a time constant of 100 ms. The input and output slit widths of the monochromator were adjusted to yield a resolution of 3–4 nm. The formal potential of the external redox couple Ferrocene/Ferrocenium (Fc/Fc$^+$) used as reference value were measured by cyclic voltammetry within the same setup and were 330 mV vs. Ag pseudo-reference electrode.

### Zeiss Axio Examiner.Z1 microscope

The images of DBOV-Mes by combined illumination of the excitation and STED beams were acquired with Zeiss Axio Examiner.Z1 with LSM 980 and Airyscan 2 (upright) microscope. 561 nm laser (continuous wave) was chosen as an excitation laser. 775 nm from SpectraPhysics InSight X3+ tunable ultrafast lasers (repetition rate 80 ± 0.5 MHz, pulse width < 120 fs) was used. We used an objective 63 × 1.40 Plan-APOCHROMAT Oil DIC.

### Home-build wide field microscope[55]

The wide field images of DBOV-Mes that reveal reactivation by 730 nm light were acquired with a home-built wide field microscope. 561 nm laser was chosen as an excitation laser with a bandpass emission filter (575–625 nm). Irradiance of about 4 kW cm$^{-2}$ was used. The laser beam was focused at the back focal plane of the objective lens (HCX PL APO 100×/NA 1.47 OIL, Leica, Germany). The emission light was focused and imaged on a sCMOS camera (PCO edge 4.2) with a pixel size corresponding to 65 nm in the sample plane. Light from a 730 nm laser (30 mW, beam diameter 0.8 mm, ≈6 W cm$^{-2}$) is reflected by a mirror and illuminates the sample vertically.

### Leica SR GSD microscope

The wide field images of DBOV-Mes that reactivation with and without 405 nm light were acquired with Leica SR GSD microscope. 532 nm (500 mW) laser was selected for excitation with a dichroic beam splitter (527–537 nm/400–410 nm) and emission filter (550–650 nm/449–451 nm). The 405 nm laser (30 mW) was selected as the back-pumping wavelength for fluorescence recovery. The objective lens HCX PL APO 160 × 1.43 NA Oil was used. The microscope was equipped with an EMCCD camera (iXonDU-897, Andor). The camera settings were 10 MHz at 14 bit and a pre-amplification of 5.1. For imaging, the camera exposure time was set to 30 ms and an EM gain of 100 was used. The pixel size of the image was 100 nm.

### Leica Stellaris 8 STED microscope

The deactivation-reactivation experiments and 2D nanographene-PS images were done with Leica Stellaris 8 system. The system has a tunable White Laser (with a range from 440 to 750 nm) and a 775 nm STED Laser (repetition rate 80 MHz). 561 nm (repetition rate 80 MHz) was selected for excitation and deactivation. We used an objective HC PL APO CS2 100×Oil.

### Leica TCS SP8 STED microscope

The 3D STED images of glass cracks were acquired with a Leica TCS SP8 STED microscope that is equipped with a white light laser source (WLL)

and 775 nm STED depletion laser. A 100× 1.4 NA HC PL APO CS2 oil-immersion objective was used.

### Abberior Expert Line microscope

All the STED images of liposomes were taken with an Abberior Expert Line microscope using a 775 nm pulsed laser (pulse duration 1.03 ns, repetition rate 40 MHz) as depletion beam and 561 nm (21 ns period) pulsed laser as excitation beam. The light was collected by a 63 × 1.4 oil objective lenses and directed, after passing through a pinhole, to an avalanche photodiode (APD) protected by red emission filter 615/20 nm.

### Image processing and data analysis

All the images were acquired on microscopes using corresponding software. All images in Figs. 1–4 and Supplementary Figs. 1, 3, 10, 13, 15–19, 24 represent raw data. Images in Fig. 5 and Supplementary Fig. 23 were processed by Richardson-Lucy deconvolution method. The images shown are not entire acquired area, but a cropped area of interest. The size of the displayed image is given by the pixels of the image and the scale bar.

The fluorescence intensity is the mean value calculated with ImageJ. For Fig. 1b–d, the fluorescence retention is the ratio (%) of the mean intensity of the deactivation area in the third and fifth steps to the intensity measured in the first step. For Fig. 2b, leftover fluorescence ratio was determined by dividing the mean intensity of each of the 150 images by the intensity of the first frame. For Figs. 2c–e and 3c, d, the mean intensity is from the deactivation area of the confocal images before and after deactivation/reactivation/imaging. It should be noted that large-size confocal images were acquired before and after deactivation/reactivation/imaging to confirm the stability of the experimental system, such that the focal plane is still maintained before and after the experiment.

### Two-photon ionization process—how molecules enter the fluorescence OFF state

We take continuous confocal images (up to 150 frames) of one imaging place with fixed excitation laser intensity (Supplementary Fig. 7). Vary the excitation laser intensity, move to a new imaging place, and repeat the imaging process.

The following is the derivation process justifying Fig. 2b of the main figures and Supplementary Fig. 8a:

$$\triangle N = - C \cdot I_{exc}^2 \cdot N \cdot \triangle t \tag{1}$$

Where $\triangle N$ is the number of nanographene molecules entering the fluorescence off state at each time step $\triangle t$; $N$ is the number of molecules staying in the fluorescence on state; $I_{exc}$ is the excitation laser intensity; $C$ is the positive rate (exponential decay constant).

From Eq. (1), we know that

$$\frac{dN}{dt} = - C \cdot I_{exc}^2 \cdot N \tag{2}$$

Therefore

$$N(t) = N_0 \cdot e^{-C \cdot I_{exc}^2 \cdot t} \tag{3}$$

And we can get

$$F_1(t) = \frac{N(t)}{N_0} = e^{-C \cdot I_{exc}^2 \cdot t} \tag{4}$$

$F_1(t) = \frac{N(t)}{N_0}$ is therefore the measured leftover fluorescence ratio of the first image.

### Data analysis method one (for Fig. 2b)

From Eq. (4), we can get

$$F_{ln}(t) = \left| \ln(F_l(t)) \right| = C \cdot I_{exc}^2 \cdot t \tag{5}$$

$$\ln(F_{ln}(t)) = 2\ln(I_{exc}) + \ln(C) + \ln(t) \tag{6}$$

From Eq. (6) we can see that the slope of $\ln(F_{ln}(t))$ vs. $\ln(I_{exc})$ should be a ratio of 2, in case of two-photon ionization.

### Data analysis method two (for Supplementary Fig. 8a)

As an alternative, people in this society are more used to characterize log-log plot of $I_{exc}$ vs. emission quenching ratio.

From Eq. 4, we can get

$$F_q(t) = 1 - F_l(t) = 1 - \frac{N(t)}{N_0} = 1 - e^{-C \cdot I_{exc}^2 \cdot t} \tag{7}$$

$F_q(t)$ is quenched fluorescence ratio. When the term $C \cdot I_{exc}^2 \cdot t$ is small (e.g., smaller than 0.1), the quenched fluorescence ratio can approximate

$$F_q(t) = 1 - e^{-C \cdot I_{exc}^2 \cdot t} \approx C \cdot I_{exc}^2 \cdot t \tag{8}$$

Therefore

$$\ln(F_q(t)) \approx \ln(C) + 2\ln(I_{exc}) + \ln(t) \tag{9}$$

With the same experimental data, a slope of 2 was calculated based on "method one" (Fig. 2b), while 1.59 was calculated based on "method two" (Supplementary Fig. 8a). A smaller slope number was achieved with method two. That is because in our experiment, the first valid data was measured at 30 frames which means longer/larger time $t$ while the laser intensity is high (at least 471 kW cm$^{-2}$). In this case, the term $C \cdot I_{exc}^2 \cdot t$ might not be small enough to make Eq. (8) valid any more. Therefore, for our data analysis, "method one" should be more accurate.

We note here: The excitation laser used in Leica stellaris 8 STED microscope setup is nanosecond pulsed laser (repetition rate 80 MHz) with an averaged laser intensity higher than 471 kW cm$^{-2}$. In this case, a simple exponential decay model (Eq. (1)) is considered.

### Single-photon reactivation process with STED/doughnut beam—how the molecules are reactivated from fluorescence OFF state back to fluorescence ON state

Experimental steps for Fig. 2c as shown in Fig. 2a(2):

Step 1: one confocal image of the selected imaging area—with a measured fluorescence intensity $F_0$;

Step 2: continuous confocal imaging of 150 frames to quench the imaging area;

Step 3: one confocal image again—with a measured fluorescence intensity $F_{deactivation}$;

Step 4: STED/doughnut beam reactivation scanning for 5 frames;

Step 5: one confocal image again—with a measured fluorescence intensity $F_{reactivation}$;

Step 6: vary the STED/doughnut beam intensity, find a new imaging area, and repeat step 1–5.

The following is the derivation process of how we having Fig. 2c of the main figures and Supplementary Fig. 8b:

$$\triangle N = -A \cdot I_{STED} \cdot N \cdot \triangle t \tag{10}$$

Where $\triangle N$ is the number of nanographene molecules entering the fluorescence ON state at each time step $\triangle t$; $N$ is the number of

molecules staying in the fluorescence OFF state; $I_{sted}$ is the STED/doughnut laser intensity which is applied in "Step 4" of the experiments described above; $A$ is the positive rate (exponential decay constant).

From Eq. (7), we know that

$$\frac{dN}{dt} = -A \cdot I_{STED} \cdot N \tag{11}$$

Therefore

$$N(t) = N_0 \cdot e^{-A \cdot I_{STED} \cdot t} \tag{12}$$

And we can get:

The ratio of the molecules in the fluorescence OFF state

$$R_{off}(t) = \frac{N(t)}{N_0} = e^{-A \cdot I_{STED} \cdot t} \tag{13}$$

### Data analysis method one (for Fig. 2c)

From Eq. (13), we can get

$$R_{ln}(t) = \left| \ln R_{off}(t) \right| = A \cdot I_{STED} \cdot t \tag{14}$$

$$\ln(R_{ln}(t)) = \ln(I_{STED}) + \ln(A) + \ln(t) \tag{15}$$

From our experimental results, we can calculate the ratio of the molecules in the fluorescence OFF state after the reactivation with STED/doughnut beam:

$$R_{off}(t) = \frac{N(t)}{N_0} = \frac{1 - \frac{F_{reactivation}}{F_0}}{1 - \frac{F_{deactivation}}{F_0}} \tag{16}$$

In Eq. (16), $F_0$, $F_{reactivation}$ and $F_{deactivation}$ are the fluorescence intensity measured in the above-mentioned experimental steps 1, 5 and 3, respectively.

From Eqs. (14–16) we can therefore calculate that the slope of $\ln(R_{ln}(t))$ vs. $\ln(I_{STED})$ from the experimental results. The slope has a ratio of 1, which confirms the single-photon reactivation process.

### Data analysis method two (for Supplementary Fig. 8b)

As an alternative, people in this society are more used to characterize log-log plot of $I_{STED}$ vs. fluorescence reactivation ratio.

From Eq. (13), we can get

$$R_{on}(t) = 1 - R_{off}(t) = 1 - \frac{N(t)}{N_0} = 1 - e^{-A \cdot I_{STED} \cdot t} \tag{17}$$

$R_{on}(t)$ is reactivated fluorescence ratio. When the term $A \cdot I_{STED} \cdot t$ is small (e.g., smaller than 0.1), the reactivated fluorescence ratio can approximate

$$R_{on}(t) = 1 - e^{-A \cdot I_{STED} \cdot t} \approx A \cdot I_{STED} \cdot t \tag{18}$$

Therefore

$$\ln(R_{on}(t)) \approx \ln(A) + \ln(I_{STED}) + \ln(t) \tag{19}$$

With the same experimental data, a slope of 1 was calculated based on "method one" (Fig. 2c), while 0.23 was calculated based on "method two" (Supplementary Fig. 8b). A much smaller slope number was achieved with method two. That is because in our experiment,

high intensity pulsed STED beam (average intensity range from 3.08 to 30.52 MW cm$^{-2}$) was used for the reactivation experiments. In this case, the term $A \cdot I_{STED} \cdot t$ might not be small enough to make Eq. 18 valid any more. Therefore, for our data analysis, "method one" should be more accurate. We note here that in this experiment, long deactivation and short reactivation time (deactivation 150 frames, pixel dwell time 102.375 μs, total collection time 203.146 s; reactivation 5 frames, pixel dwell time 102.375 μs, total collection time 5.454 s) was used to avoid saturation reactivation effect. Under different reactivation beam intensities, more fluorescence can be recovered when the reactivation time is increased.

## Data availability
The data that support the findings of this study are available from Figshare[56] and from the corresponding authors upon request. Source data are provided with this paper.

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

## Acknowledgements

We acknowledge the Light Microscopy Core Facility at Johannes Gutenberg University and especially Christof Rickert and Bastian Hülsmann for introduction into the Leica Stellaris 8 STED microscope and assistance. We appreciate Shih-Ya Chen for help with the measurements (reactivation by 405 nm light and spontaneously decay). We thank Sandra Ritz (IMB Microscopy Core Facility) for technical support. We acknowledge Nikita Kavokine (Max Planck Institute for Polymer Research), Johan Hofkens (KU Leuven), Allard Mosk (Utrecht University), Thomas Basché (Johannes Gutenberg-Universität Mainz) and Christoph Cremer (Max Planck Institute for Polymer Research) for discussions. This work was financially supported by the Max Planck Society and the Okinawa Institute of Science and Technology Graduate University. Q.Y. and X.Z are recipients of scholarship from China Council Scholarship (CSC). The utilization of Zeiss Axio Examiner.Z1 microscope was supported by DFG major instrumentation funding (INST 152/876-1 FUGG). The funding for the Leica Stellaris 8 STED instrument was provided by the DFG INST 247/1004-1 FUGG. M.S. acknowledges the DFG for funding under grant SCHE1905/9-1 (project no. 426008387). G. and H.Z. appreciates the JSPS Postdoctoral Fellowships for Research in Japan.

## Author contributions

X.L., A.N., M.B., and R.K. initiated and supervised the entire project. R.K. proposed the fluorescence deactivation and reactivation mechanism. Q.Y. performed the daily project supervision, and planned and performed most of the experiments and data analysis with input from A.V.F., W.Y., P.T., W.Z., and M.G. Q.Y. and A.V.F. recorded the STED images of liposomes; P.T. and Q.Y. performed materials STED imaging and initial STED reactivation experiments; liposomes were provided by A.M.M., M.G., S.M. and K.L.; long term 3D materials STED imaging was performed by W.Z. and X.L.; EMAS experiments were conducted by S.W. and M.S.; M.G. and W. Y. helped with data analysis; Q.C., G. and H.Z. synthesized nanographene DBOV-Mes; X.Z. helped for the preparation of liposomes for imaging. Q.Y., X.L., A.N., M.B., and R.K. co-wrote this manuscript with input from all authors.

## Funding

## Competing interests

This work is related to two patents. For patent 1 (PCT/EP2019/076496, WO2020070085): Patent applicant is the institution Max Planck Society. X.L., A.N., Sapun Parekh, Q.C., Klaus Müllen, Christoph Cremer, K.L., M.B. are named as inventors. Status of the application: EP3633009A1 priority application filed 03.10.2018 is lapsed; EP2018199451A1 priority application filed 09.10.2018 is lapsed; WO2020070085A1 filed 30.09.2019 was nationalized in EP, US, CN; EP3830217B1 granted, validated in BE, CH, DE, FR, GB, NL; CN113330096B granted; US20210388260A1 under examination. Specific aspect of manuscript covered in patent application: the use of a substituted or unsubstituted polycyclic aromatic hydrocarbon compound for high-resolution microscopy methods including STED. For patent 2: Patent applicants are the institutions Max Planck Society and Okinawa Institute of Science and Technology. Q.Y., R.K., M.B., A.N., and X.L. are named as inventors. Status of application: The unpublished EP priority application was filed on 22.03.2024 and a search is requested. The EP patent application was filed at the European Patent Office in Munich, Germany. Specific aspect of manuscript covered in patent application: a method for fluorescence recovery properties of nanographene described in this study. Apart from this, the authors declare no competing interests.
