## [Transparent Peer Review file · Nature Communications]

Reactivatable stimulated emission depletion microscopy using fluorescence-recoverable nanographene

Corresponding Author: Professor Mischa Bonn

Version 0:

Reviewer comments:

Reviewer #1

(Remarks to the Author)

This is an interesting article in which a molecular graphene or nanographene based on a ovalene derivative, DBOV-Mes, is used to show a new optical application such as in STED microscopy. An improvement of the STED resolution is described which is notable. The new aspect of this investigation is that the STED beam acts simultaneously as a fluorescence recoverable agent at the same time that did its work on increasing resolution by beam stimulated emission depletion. Since STED applications of organic fluorophores is known and the lasing properties of DBOV-Mes already reported by the same authors and others, the novelty of the work resides on the recoverable emission action of the STED beam. The subsequent optical characterization of the optical properties and the interesting applications in biological liposome media are very appealing, I must recognize this is not my main field of research so I assume this is correct.

My main reviewer evaluation goes with the photophysical mechanism. This has several points.

1) The exciting mean at 561 nm produces by multiphoton absorption radical anions and cation by ionization that do not degrade further within the area of excitation. Because these do not further react, stimulated recombination of charges is fueled that recovers the neutral non-degraded ground state. I would like to see the behavior in presence of oxygen which degrades the radicals and makes unviable the stimulated charge recombination recovery. This can be a counterproof of the mechanism.

2) DBOV-Mes has been studied in non-linear optical applications. I would like to hear the authors about the two photon excitation fluorescence. Why not the STEP beam at 775 nm might further activate the fluorescence emission by two photon absorption instead of mainly by stimulated charge recombination?.

3) The authors assume that once the radical cation is generated, absorption of 775 nm light increase its electron acceptor character in the excited state that accelerates the oxidation of the nearby anion going back both to the neutral state. Right? I would like to see the absorption spectrum of the radical anion of DBOV-Mes. It is known that radical anions and cations of these NG systems have similar optical properties and thus would like to see that the 775 nm excitation pumps the radical cation and not the radical anion.

4) Since the mechanism of fluorescence recovery by the STEP beam is based on the charge recombination of cations and anions, I would like to see a proof of the bimolecular nature of this process, for instane, with varying concentrations of the sample.

Just curious, why the authors name REMI to ionization by resonance enhanced multiphoton ionization when this is known for quite long time by REMPI. Please clarify or correct.

Responses and manuscript changes taking into account these issues have to be considered before a final decision is made.

Reviewer #2

(Remarks to the Author)

Review of MS „ReSTED: Reactivatable stimulated emission depletion microscopy using fluorescence-recoverable nanographene”

In the present manuscript, the authors describe the photophysical properties of the nanographene dibenzo[hi,st]ovalene

(DBOV) that render it an ideal fluorophore for long-term imaging using STED microscopy. They do so using multiple complementary approaches ranging from imaging the nanographene adsorbed to microscope slides to imaging biomimetic liposomes labeled with DBOV. Thereby, they show that exposure of the fluorophore to high levels of near IR laser illumination, as is typically done during a STED imaging session, robustly regenerates the nanographene's capacity to fluoresce after getting de-activated through the specific excitation at lower wavelengths. They further show that this behavior of DBOV is independent of the fluorophore's chemical environment. In particular, they show that it is independent of the presence of chemical anti-fading reagents. The authors conclude that these properties of DBOV will contribute to major improvements in STED imaging, allowing for more prolonged imaging sessions without bleaching than most of the currently used organic fluorophores in STED. They point out that especially the independence of antifading agents will widen the potential uses of STED microscopy in materials sciences and live cell applications in the life sciences.

The supplemental methods and notes are extensive and very detailed. This is especially true for the level of detail the authors provide on the parameters of digital imaging that they used in each of the microscopy experiments. This level of detail should enable colleagues in the field to reproduce the experiments. I commend the authors for this effort.

This reviewer thinks that this is an excellent manuscript describing carefully planned and conducted work. The results are well documented and illustrated. They support the conclusions presented by the authors. The findings in this manuscript have the potential to be transformative for a burgeoning field in microscopy at the forefront of multiple disciplines in the life and material sciences, especially so if they manage in further studies to tether the DBOV to antibodies or nanobodies. I only have a few minor concerns that need to be addressed to make the manuscript suitable for publication. I will detail them below.

Minor concerns:

Supplementary Fig. 2/Supplementary Methods: In the legend to Suppl. Fig. 2 the authors state that the data were collected using a home-built wide field microscope. They state that "532 nm laser was used for excitation". However, in the Methods section pertaining to this microscope it is described as possessing a 561 nm laser for excitation. Please clarify and correct.

Figure 1B: In Figure 1 B, discussed in the 1st paragraph of p. 2 of the manuscript, the authors show that in a region of interest previously deactivated by confocal scanning, the fluorescence is recovered by subsequent scanning with a 775 nm STED depletion laser. The recovery is qualitatively illustrated in the third panel of Fig. 1B. The figure would gain by adding a quantification to the recovery, for instance by showing the intensity profile along a line drawn through the ROI.

Supplementary Figure 6: The visualizations of the calculated 2D STED intensity distributions in x,y and x,z in this figure are tiny. This is really a minor comment, but could they be enlarged a bit for the benefit of the reader?

Page 4, ll. 35-37 / Fig. 1B: The authors observe in their reactivation experiments that in many of their measurements the fluorescence intensity in their ROI is higher in the second confocal image, taken after scanning the ROI with the depletion/reactivation beam, than in the first confocal image taken at the start of the measurement. They explain that this may be "because additional beam excitation is required to find the focal plane and select the target imaging area before imaging" (cf. Fig. 1B). I would suggest including fiducial markers in the sample that enable finding the focal plane without exciting the DBOV prior to the measurement. This should alleviate that phenomenon. Such an experiment should be carried out. It could easily show whether the authors' explanation is right or whether another phenomenon may be at play here.

Page 8, l. 27 – typo: Please write "efficient" in place of "efficiently".

Reviewer #3

(Remarks to the Author)

A critical issue in STED microscopy is the propensity of typically used fluorophores to bleach due to the high laser intensities required for this technique. This is particularly problematic for 3D STED or timelapse STED where repeated imaging of the same area is required. The ability to reactivate bleached fluorophores represents a significant and exciting development in this field.

In this work, the authors demonstrate the potential of nanographene-based fluorophores to address this issue. Overall, this is a nice piece of work, but it requires some revisions and clarifications to confirm the conclusions and to highlight the applicability of these fluorophores to the areas that most require these capabilities e.g. timelapse STED imaging of live biological samples.

Specific Comments:

Page 2 lines 35-38 and Supplementary Figure 1

The bleaching step in between the two confocal images only exposed the sample to the 775 nm depletion laser and not to the 561 nm excitation laser. This is not representative of an actual long-term STED imaging experiment, and fluorophores that are not excited would not get bleached, unless the STED beam itself is able to excite the fluorophore. If this is aiming to show the effects on DBOV-Mes of the 775 nm laser alone, it should perhaps also include a discussion of the presence or

absence of excitation of DBOV-Mes at 775 nm (also see comment about page 6 lines 14-15 and Supplementary Figure 9).

Page 4 lines 3-8, 28-33 and Figure 1B

The explanation about the reactivation of the entire area of ROI1 despite only scanning ROI2 should be moved from lines 28-33 to directly after line 8 to aid with understanding of this result.

Supplementary Figure 4

Define what the red and black arrows show.

Page 6 lines 14-15 and Supplementary Figure 9

How was it determined that this was an activation resulting from a two-photon effect of the STED beam and not from the STED beam catching the tail end of DBOV-Mes's excitation spectrum? It is earlier stated that the STED laser "does not coincide with the absorption range" (page 2 lines 31-32). Was this confirmed experimentally?

Supplementary Figure 11

Clarify if step 2 includes scanning with the 775nm and 561 nm lasers, or 775 nm laser alone.

Page 6 lines 36-39 and Figure 3B

Given the around 100% fluorescence retention in Fig2D when only a few deactivation frames were acquired before reactivation, what explains the rapid drop in fluorescence retention during STED imaging? Would you not expect that deactivation and reactivation would be occurring rapidly within each frame so it should stay at around 100% retention?

Figure 4

It would be good to compare the achievable improvement in resolution to a comparable dye that is optimised for STED e.g. Abberior STAR RED. Likewise, it would also be helpful to show the resolution of the same structure imaged repeatedly with STED to see if there is a degradation of resolution in later images.

Page 9 line 17

It does not seem accurate to refer to reconstruction of 2D-STED images into a 3D volume as 3D-STED if there is no improvement in Z resolution. 3D-STED should refer to imaging where there is depletion in both XY plane and the Z-axis.

Discussion

The authors refer to examples in the literature showing nanographene biocompatibility etc. which helps to highlight the potential applications of DBOV-Mes. However, they could also include reference to any capabilities to functionalise nanographenes for targeting specific structures of interest.

It would also be of significant interest for the authors to discuss the possibility of combining nanographenes with different spectral properties for multi-coloured STED as this would greatly improve the usefulness of these for imaging of biological samples.

Note

The photo-chemistry included in this manuscript is outside of my area of expertise and I am therefore unable to comment on this.

Reviewer #4

(Remarks to the Author)

Yang and colleagues report on the use of a red-emitting nanographene, DBOV, in STED microscopy. They discover that DBOV is photostable under STED conditions. Following signal reduction during light exposure, a large portion of fluorescence can be recovered by infrared light exposure. Only with prolonged intense light exposure does a permanent degradation of the signal occur. Hence they term their approach reactivatable STED. "Non-bleaching" or reactivatable fluorophores would be a very valuable addition to the STED toolbox. As such, the study is relevant and timely. The current study is mostly a photophysical characterization of nanographene deactivation and reactivation on a STED microscope. My major point of criticism is that the actual STED imaging performed in the study does not seem to approach the current state of the art performance with organic fluorophores. STED imaging is performed on technical samples (cracks on coverslips, DBOV embedding in polymer) and liposomes, which may point to a difficulty in functionalizing the nanographenes for more general imaging applications. The achieved resolution and – as far as this can be judged from the data provided – signal-to-noise ratio appear lower than with state-of-the-art fluorophores. It should also be discussed what the limitations in resolution and signal to noise ratio are. If photobleaching is not limiting, what determines/limits the quality of the STED images? To make a real difference, ideally, the work would be complemented by high-performance STED measurements in biological samples, including e.g. antibody labeling or live imaging highlighting specific cellular molecules, as this seems to be the promise put forth by the authors. With its current scope, the work may thus be more suited for a more specialized chemistry/biophysics journal.

Minor points

- It would be helpful to exactly describe how individual measurements were performed in figure captions and define all quantities displayed also there.
- Abstract: "tradeoff with imaging time" is slightly confusing. It's more the number of images that can be acquired.

- p1, l39: Doughnut shape is not essential to STED, many other light patterns can be used.
- p2, l6: “high excitation powers”. It’s not just excitation power but also importantly STED laser power that induces photobleaching.
- p2, 35: “photon bleaching” should be “photo bleaching”
- p4, l35: Phrasing may be adjusted: Some molecules may also be in the deactivated state at equilibrium. This may potentially be due to ambient light exposure, not just due to the excitation light exposure during focus finding.
- Data in Fig. 5 BD are deconvolved. Rather show original data.
- Suppl. Fig. 2: It should be stated in the figure caption what imaging buffer was used. Alexa 647 does not seem to be a particularly photostable conventional fluorophore to compare to.

Version 1:

Reviewer comments:

Reviewer #1

(Remarks to the Author)

After reading the responses to my concerns and new experiments/information/discussion from the authors in the new version of the article, I am satisfied either with the responses to my criticisms as with the changes in the revised version of the manuscript.

I also care of the responses to the others reviewers, and certainly, the work of the authors is mostly convincing to me as well.

In these circumstances, I am in favor of acceptance of the present version of the article in Nat. Comm.

Reviewer #2

(Remarks to the Author)

As already stated in my original review of this manuscript, I think it is an excellent piece of work, which has the potential to make a big impact on a burgeoning field in advanced microscopy, namely STED microscopy. In the present, revised version, I find the authors' responses to the reviewers' comments extensive, including new experiments, and predominantly convincing. Importantly, they convinced me that DBOV-OPEG-N3 can be conjugated to a peptide, which enables it to enter the cytoplasm of the root cells of a plant. This success opens the route for the graphene to become a bona fide tool in the hands of cell biologists who need more photostable fluorophores in long-term live cell experiments. Unfortunately, this conjugation also induces a red shift in the graphene's emission, which renders the conjugate non-depletable with the depletion lasers currently available to the authors. For this reason, they resort to showing us confocal images to document their achievement. This is in itself enough for me to accept this additional experiment, although the images only convince me that they got the conjugated graphene into the cells, not really that it labels mitochondria. But that does not take away enough of the overall excellence and impact of the work to request more experiments at this point.

Reviewer #3

(Remarks to the Author)

Overall, I am very satisfied with how the authors addressed the questions raised in my initial review. There is one point that was not fully addressed - the question of comparing achievable resolution with a known STED fluorophore such as STAR RED. The authors added in a comparison of bleaching but not of resolution. However, I understand that this would require significant additional work to label the same structure with both dyes so I am happy to recommend that the revised paper be accepted.

Reviewer #4

(Remarks to the Author)

I appreciate the additional work invested by the authors. I point out that the manuscript is mostly a photophysical study, rather than an actual demonstration of improved performance in practically applicable STED microscopy. Suitability for the journal is ultimately at the editor's discretion.

Below please find specific comments on the revised manuscript.

1) Original comment:

My major point of criticism is that the actual STED imaging performed in the study does not seem to approach the current state of the art performance with organic fluorophores. STED imaging is performed on technical samples (cracks on coverslips, DBOV embedding in polymer) and liposomes, which may point to a difficulty in functionalizing the nanographenes for more general imaging applications.

Authors' response:

We thank the reviewers for this very useful comment. When evaluating the performance of actual STED imaging, several key metrics are usually considered: spatial resolution, signal to noise ratio (SNR), photobleaching, imaging time, laser power, uniformity, depth of field (DOF), spectral crosstalk for multicolor imaging, etc. In this work, we focus on two key metrics, photobleaching and imaging time: 1) our demonstration of nanographene for ReSTED imaging gets rid of the dependence of

currently used organic fluorophores on anti-fading agents; and 2) long-term 3D STED imaging time is achieved. From these two points of view, the imaging technology we demonstrated is superior to the state-of-the-art performance of organic fluorophores. In long term studies, we will further investigate other key metrics that go beyond the current work. Regarding the resolution and signal-to-noise ratio pointed out by the reviewer, we have provided further discussion in our response to “Concern 2” below.

Comment on response:

I agree that there are several performance metrics relevant to STED microscopy. I do however point out that these are highly interlinked. E.g. photobleaching strongly depends on the desired resolution. As pointed out above, I thus argue that the present study is an interesting demonstration of photophysics rather than a concrete advance for practical STED imaging.

2) Original comment:

The achieved resolution and – as far as this can be judged from the data provided – signal-to-noise ratio appear lower than with state-of-the-art fluorophores. It should also be discussed what the limitations in resolution and signal to noise ratio are. If photobleaching is not limiting, what determines/limits the quality of the STED images?

Authors' response:

We thank the reviewer for the insightful comments.

The resolution in STED microscopy is determined by several parameters and can be described by the following formula:

$$d = \lambda / (2 \cdot NA \cdot \sqrt{1 + I_{\text{STED}} / I_{\text{sat}}}) \quad (1)$$

Where d is the achievable resolution, λ is the STED beam wavelength, NA is the numerical aperture of the objective lens, I_{STED} is the intensity of the STED beam, I_{sat} is the saturation intensity of the fluorophore, which can be calculated by $I_{\text{sat}} = hv / (\sigma \tau_{\text{FL}})$ (2)

Where $h\nu$ is the photon energy at the STED wavelength, σ is the cross-section for stimulated emission, τ_{FL} is the fluorescence lifetime of the fluorophore (Nature Photonics 3, 144–147 (2009); Opt. Express 19, 8066-8072 (2011)).

The stimulated emission cross section σ has a spectral dependency,

$$\sigma(\lambda) = (\lambda^4 E(\lambda) \phi) / (8 \pi c n^2 \tau_{\text{EL}}) \quad (3)$$

Where λ is the wavelength, $E(\lambda)$ is the normalized emission spectrum at wavelength λ , ϕ is the fluorescence quantum yield of the fluorophore, c is the speed of light, n is the refractive index, and τ_{EL} is the excited state lifetime (Nanomaterials 2022, 12(1), 21).

From Eq. (1), it can be seen that among all the parameters affecting the imaging resolution, only the saturation intensity I_{sat} of the fluorophore is an intrinsic property of the fluorophore, while the other parameters are determined by the microscope imaging setup. To achieve high resolution, a lower I_{sat} is desirable for the fluorophore. Furthermore, as shown in Eq. (2), obtaining a low I_{sat} requires a long fluorescence lifetime τ_{FL} , and a high stimulated emission cross-section.

In our study of the nanographene DBOV-Mes, the measured fluorescence lifetime is around 3 ns, comparable to that of the other organic dyes. Regarding the stimulated emission cross-section, several factors play a role (as shown in Eq. (3)): the fluorescence quantum yield ϕ , the refractive index n , and the excited state lifetime τ_{EL} , all of which are intrinsic properties of the fluorophore. The typical refractive index n for common organic dyes and carbon dots is around 2 (~1.5-2.4) (Nanoscale, 2022, 14, 8145-8152; RSC Adv., 2017, 7, 36632-36643; Polymers 2021, 13(15), 2545). Notably, Nanographene DBOV-Mes exhibits a very high fluorescence quantum yield of ~80% and a fast excited state lifetime of ~0.5 ps (Mater. Horiz., 2022, 9, 393-402). Consequently, DBOV-Mes is expected to have a high stimulated emission cross-section, which corresponds to a low saturation intensity. Therefore the resolution imaged with nanographene DBOV-Mes should be comparable to the state of the art of organic molecules.

Comment on response:

The authors recapitulate the well-known square root dependence of STED resolution on intensity, with the saturation intensity setting the scale. It would be informative and substantially strengthen the manuscript to actually measure the saturation intensity and/or give resolution measurements that are on a par with the resolution achieved with organic fluorophores.

Authors' response:

The signal to noise ratio (SNR) quantifies the level of the desired signal in relation to background noise. To achieve a higher signal, it is preferable to use bright fluorophores. As noted in reference (Angew. Chem. Int. Ed. 2020, 59, 496–502), DBOV-Mes has a brightness of 55300 M⁻¹cm⁻¹, comparable to standard organic dyes like Alexa 647.

Additionally, to maximize fluorescence signal detection, it is crucial to select an appropriate excitation laser wavelength and an emission bandpass filter that aligns with the fluorophore's emission spectrum. Currently, only the commercial Abberior Expert Line microscope available in our laboratory supports true 3D STED imaging. However, this setup utilizes a fixed emission filter of 605-625 nm for red dyes, which unfortunately filters out approximately 68% of the fluorescence signal from our nanographene DBOV-Mes. To enhance the SNR, a tunable emission filter that can be adjusted to match the emission spectrum of DBOV-Mes would be ideal.

As we above described in the response to “Concern 1”, discussing the limitations in resolution and SNR is not the primary focus of the current work. However, to provide relevant information to readers, we have included the following sentences in the revised manuscript:

“Notably, while DBOV-Mes is theoretically capable of achieving high resolution under intense STED laser, it is essential to use an appropriate tunable emission filter for enhancing the signal-to-noise ratio (SNR).”

Comment on response:

For collecting more of the emitted fluorescence, it is not necessary to have a tunable filter. Merely exchanging the detection bandpass filter would be sufficient, which is easily possible in the Abberior Expert Line microscope. The authors point to a limitation for 3D STED imaging. 3D STED imaging is not necessary for measuring lateral resolution, which would be sufficient in this case.

3) Original Comment:

To make a real difference, ideally, the work would be complemented by high-performance STED measurements in biological samples, including e.g. antibody labeling or live imaging highlighting specific cellular molecules, as this seems to be the promise put forth by the authors.

Comment:

The authors indicate that STED measurements are not possible with current biocompatible variants due to a spectral shift. I appreciate this limitation.

Further comments:

Supplementary Fig. 1:

„The fluorescence intensity after scanned by the STED beam was even higher than in the first image because additional beam excitation is required to find the focal plane and select the target imaging area before imaging. Therefore, before taking the first confocal image, some DBOV-Mes molecules may already be in the fluorescent OFF state.“
I deem it equally likely that there is a fraction of molecules in a non-fluorescent state in equilibrium. This population may get activated by the STED beam, resulting in higher intensity after scanning with the STED beam.

Supplementary Fig. 2:

Are photobleaching properties of Abberior STAR RED after evaporation of the solution similar as in the more usual setting of aqueous solution?

Reviewer #1:

This is an interesting article in which a molecular graphene or nanographene based on a ovalene derivative, DBOV-Mes, is used to show a new optical application such as in STED microscopy. An improvement of the STED resolution is described which is notable. The new aspect of this investigation is that the STED beam acts simultaneously as a fluorescence recoverable agent at the same time that did its work on increasing resolution by beam stimulated emission depletion. Since STED applications of organic fluorophores is known and the lasing properties of DBOV-Mes already reported by the same authors and others, the novelty of the work resides on the recoverable emission action of the STED beam. The subsequent optical characterization of the optical properties and the interesting applications in biological liposome media are very appealing, I must recognize this is not my main field of research so I assume this is correct.

We thank the reviewer for the thorough and very supportive review of our work. In the following, we provide a point-by-point reply to the comments raised by the reviewer.

My main reviewer evaluation goes with the photophysical mechanism. This has several points.

1) The exciting mean at 561 nm produces by multiphoton absorption radical anions and cation by ionization that do not degrade further within the area of excitation. Because these do not further react, stimulated recombination of charges is fueled that recovers the neutral non-degraded ground state. I would like to see the behavior in presence of oxygen which degrades the radicals and makes unviable the stimulated charge recombination recovery. This can be a counterproof of the mechanism.

This study was also conducted under air (presence of oxygen), and similar fluorescence recovery was observed (Fig. 3). Therefore, it is considered that quenching of radical cations by oxygen has a low probability. This is likely because the HOMO of the DBOV-Mes radical cation is deep, resulting in a low reactivity with oxygen.

In the presented mechanism, DBOV-Mes is first ionized by resonance enhanced multiphoton ionization (REMPI), producing a DBOV-Mes radical cation and a free electron. If the concentration of DBOV-Mes is high, there is a possibility that a nearby DBOV-Mes could receive the electron and form a DBOV-Mes radical anion. However, the ReSTED is observed even under the very low concentration that each DBOV-Mes is well isolated conditions (more detailed discussion in response for comment 4). Therefore, the electron acceptor does not necessarily have to be DBOV-Mes.

During the stimulation process, the photo absorption of the DBOV-Mes radical cation increases its electron acceptor properties, which accelerates the oxidation of the nearby anion or other electron sources, returning to the neutral state as the reviewer pointed out in comment 3.

2) DBOV-Mes has been studied in non-linear optical applications. I would like to hear the authors about the two photon excitation fluorescence. Why not the STEP beam at 775 nm might further

activate the fluorescence emission by two photon absorption instead of mainly by stimulated charge recombination?

We thank the reviewer for this thoughtful comment. We have also considered the possibility that the STED beam at 775 nm could further activate two-photon fluorescence emission when the beam peak intensity is high enough, since pulsed laser beam at 775 nm is applied in our used microscope setup. To investigate this, we conducted measurements at different STED laser power densities. At low laser power density, we did not observe notable two-photon fluorescence. However, at higher laser power density, two-photon fluorescence was evident (Supplementary Figure 9). This indicates that the two-photon fluorescence activation may vary depending on the laser power density.

Moreover, we also tested the fluorescence retention after different STED power densities scanning on DBOV-Mes. After prolonged scanning (up to 100 frames, as shown in Supplementary Fig. 11), the fluorescence decreasing was not significant when scanned with 50% STED power (140 mW) compared to 100% STED power (270 mW). This might be induced by the two-photon effect.

In order to address this, we have the sentences '*two-photon fluorescence was observed when increasing the reactivation beam peak intensity (Supplementary Fig. 9) ...*' and '*...To address these issues, one solution could be the use of longer pulse duration or even continuous wave (CW) reactivation beam...*' on page 6 in our main text.

We now modified '*two-photon fluorescence was observed when increasing the reactivation beam peak intensity (Supplementary Figs. 10 and 11) ...*' of the revised main text and added the following figure in the revised supporting information:

Supplementary Figure 11: fluorescence retention after different STED power densities scanning

Measured leftover fluorescence ratio over time (STED beam scanned frames). Left: 100% STED power (270 mW measured at objective) was set. Right: 50% STED power (140 mW measured at objective) was set. Experimental details are same as described in Supplementary Figure 1.

3) The authors assume that once the radical cation is generated, absorption of 775 nm light increase its electron acceptor character in the excited state that accelerates the oxidation of the nearby anion going back both to the neutral state. Right? I would like to see the absorption spectrum of the radical anion of DBOV-Mes. It is known that radical anions and cations of these

NG systems have similar optical properties and thus would like to see that the 775 nm excitation pumps the radical cation and not the radical anion.

Text Redacted

Figure Redacted

4) Since the mechanism of fluorescence recovery by the STEP beam is based on the charge recombination of cations and anions, I would like to see a proof of the bimolecular nature of this process, for instance, with varying concentrations of the sample.

We thank the reviewer for the suggestion regarding the proof of the bimolecular nature of the fluorescence recovery mechanism. To address this, we conducted experiments with varying concentrations (from μM to nM) of the sample to observe the effect on fluorescence recovery. At nM concentration, single-molecule level at some imaging areas were observed and the NIR light could still reactivate the fluorescence.

We now added '*We note here that the reactivation is not sensitive to the concentration of DBOV-Mes molecules (Supplementary Fig. 17).*' into page 8 of the revised main text and the following figure in the revised supporting information:

Supplementary Figure 17: Reactivation by NIR light at 730 nm

Images acquired from Home-build widefield microscope. Widefield images of DBOV-Mes before (left) and after (middle) deactivation and after reactivation by 730 nm light (right). Upper: DBOV-Mes in cracks of coverslip (μM); lower: DBOV-Mes on coverslip surface (nM). Scale bars, 5 μm .

Just curious, why the authors name REMI to ionization by resonance enhanced multiphoton ionization when this is known for quite long time by REMPI. Please clarify or correct.

We appreciate the reviewer to pointing this out and apologize for the confusion. We have corrected the abbreviation for resonance enhanced multiphoton ionization to REMPI.

Responses and manuscript changes taking into account these issues have to be considered before a final decision is made.

Reviewer #2:

Review of MS „ReSTED: Reactivable stimulated emission depletion microscopy using fluorescence-recoverable nanographene”

In the present manuscript, the authors describe the photophysical properties of the nanographene dibenzo[hi,st]ovalene (DBOV) that render it an ideal fluorophore for long-term imaging using STED microscopy. They do so using multiple complementary approaches ranging from imaging the nanographene adsorbed to microscope slides to imaging biomimetic liposomes labeled with DBOV. Thereby, they show that exposure of the fluorophore to high levels of near IR laser illumination, as is typically done during a STED imaging session, robustly regenerates the nanographene’s capacity to fluoresce after getting de-activated through the specific excitation at lower wavelengths. They further show that this behavior of DBOV is independent of the fluorophore’s chemical environment. In particular, they show that it is independent of the presence of chemical anti-fading reagents. The authors conclude that these properties of DBOV will contribute to major improvements in STED imaging, allowing for more prolonged imaging sessions without bleaching than most of the currently used organic fluorophores in STED. They point out that especially the independence of antifading agents will widen the potential uses of STED microscopy in materials sciences and live cell applications in the life sciences.

The supplemental methods and notes are extensive and very detailed. This is especially true for the level of detail the authors provide on the parameters of digital imaging that they used in each of the microscopy experiments. This level of detail should enable colleagues in the field to reproduce the experiments. I commend the authors for this effort.

This reviewer thinks that this is an excellent manuscript describing carefully planned and conducted work. The results are well documented and illustrated. They support the conclusions presented by the authors. The findings in this manuscript have the potential to be transformative for a burgeoning field in microscopy at the forefront of multiple disciplines in the life and material sciences, especially so if they manage in further studies to tether the DBOV to antibodies or nanobodies. I only have a few minor concerns that need to be addressed to make the manuscript suitable for publication. I will detail them below.

We thank the reviewer for the thorough and very supportive review of our work.

Text Redacted

Figure Redacted

Minor concerns:

Supplementary Fig. 2/Supplementary Methods: In the legend to Suppl. Fig. 2 the authors state that the data were collected using a home-built wide field microscope. They state that “532 nm laser was used for excitation”. However, in the Methods section pertaining to this microscope it is described as possessing a 561 nm laser for excitation. Please clarify and correct.

We apologize for the possible confusion that might be caused in the description of home-built wide field microscope. We have used two home-built wide field microscope setups, one is equipped with 532 nm laser and another is equipped with 561 nm laser. The home-built wide field microscope with 532 nm laser was only used to compare the photostability of DBOV with Alexa 647 as we discussed in the legend of Suppl. Fig. 2. So that we did not put this setup into Supplementary Methods part. As another reviewer suggested, it’s better to compare the photostability with a comparable dye that is optimised for STED e.g. Abberior STAR RED. Such new experiment was done with a Leica Stellaris 8 STED microscope so that we deleted the description of the home-built wide field microscope with 532 nm laser. We have replaced Supplementary Fig. 2 in revised supplementary as follows.

Supplementary Figure 2: Photo-bleaching properties of DBOV by excitation laser

Photo-bleaching properties of DBOV-Mes and STAR RED as a function of the imaging time. Samples were dyes dropped on coverslips and measured in air after the solution evaporated. The measurements were performed with a Leica Stellaris 8 STED microscope. Series of Confocal images were acquired for comparison of photo-bleaching properties. Confocal images (500 frames) were acquired using an excitation wavelength of 610 nm at 20% laser power. Emission was collected across a range from 620 nm to 750 nm. Setting: 64 pixel X 64 pixel, 0.159 $\mu\text{m}/\text{pixel}$, pixel dwell time 102.375 μs , total collection time is 340.059 s.

Figure 1B: In Figure 1 B, discussed in the 1st paragraph of p. 2 of the manuscript, the authors show that in a region of interest previously deactivated by confocal scanning, the fluorescence is recovered by subsequent scanning with a 775 nm STED depletion laser. The recovery is qualitatively illustrated in the third panel of Fig. 1B. The figure would gain by adding a quantification to the recovery, for instance by showing the intensity profile along a line drawn through the ROI.

We thank the reviewer's suggestion and have added line profiles to directly show the fluorescence change before and after deactivation and after reactivation. We have added one sentence in revised main text as "*Representative line profiles clearly show the intensity change (Supplementary Fig. 3).*" and also a figure in revised supplementary as follows.

Supplementary Figure 3: Representative line profiles from images in Fig. 1B

A-C, Confocal images shown before deactivation, after deactivation, and after reactivation as depicted in Fig. 1B. D, Representative line profiles corresponding to the yellow lines marked in the images.

Supplementary Figure 6: The visualizations of the calculated 2D STED intensity distributions in x,y and x,z in this figure are tiny. This is really a minor comment, but could they be enlarged a bit for the benefit of the reader?

We appreciate the reviewer's suggestion and have added enlarged figures of the central part to Supplementary Figure 7 in revised supplementary information.

Supplementary Figure 7: Calculated 2D STED intensity distributions

Calculated 2D STED intensity distributions in xy direction (A and C) and xz direction (B and D). Lower panels (C and D) are the enlarged views of the central part of the upper panels (A and B). Color bars show the intensity in MW/cm², maximum value was set to 200 MW/cm².

Page 4, ll. 35-37 / Fig. 1B: The authors observe in their reactivation experiments that in many of their measurements the fluorescence intensity in their ROI is higher in the second confocal image, taken after scanning the ROI with the depletion/reactivation beam, than in the first confocal image taken at the start of the measurement. They explain that this may be “because additional beam excitation is required to find the focal plane and select the target imaging area before imaging” (cf. Fig. 1B). I would suggest including fiducial markers in the sample that enable finding the focal plane without exciting the DBOV prior to the measurement. This should alleviate that phenomenon. Such an experiment should be carried out. It could easily show whether the authors’ explanation is right or whether another phenomenon may be at play here.

We thank the reviewer for the valuable suggestion. We did consider this issue, and in our experiments, we used a gridded coverslip (<https://ibidi.com/coverslips-and-foils/205-gridded-glass-coverslips-grid-50.html>). The imprinted grid structure can be seen under bright field to

determine the focal plane. However, bright field imaging cannot provide an exact focal plane, and we always need to use the excitation beam to locate the focal plane precisely.

Additionally, we prepared samples with fluorescent beads (0.1 μm TetraSpeck™ microspheres) and attempted to use deep red light to locate the beads and achieve precise focusing. However, there is still unavoidable excitation of the nanographene due to its weak absorption in the 650–700 nm range.

Page 8, l. 27 – typo: Please write “efficient” in place of “efficiently”.

We appreciate the reviewer for pointing this out. We have corrected “*efficiently*” to “*efficient*”.

Reviewer #3 (Remarks to the Author):

A critical issue in STED microscopy is the propensity of typically used fluorophores to bleach due to the high laser intensities required for this technique. This is particularly problematic for 3D STED or timelapse STED where repeated imaging of the same area is required. The ability to reactivate bleached fluorophores represents a significant and exciting development in this field.

In this work, the authors demonstrate the potential of nanographene-based fluorophores to address this issue. Overall, this is a nice piece of work, but it requires some revisions and clarifications to confirm the conclusions and to highlight the applicability of these fluorophores to the areas that most require these capabilities e.g. timelapse STED imaging of live biological samples.

We are grateful to the reviewer for carefully reading and constructively commenting our manuscript. In the following, we provide a point-by-point reply to the comments raised by the reviewer.

Specific Comments:

Page 2 lines 35-38 and Supplementary Figure 1

The bleaching step in between the two confocal images only exposed the sample to the 775 nm depletion laser and not to the 561 nm excitation laser. This is not representative of an actual long-term STED imaging experiment, and fluorophores that are not excited would not get bleached, unless the STED beam itself is able to excite the fluorophore. If this is aiming to show the effects on DBOV-Mes of the 775 nm laser alone, it should perhaps also include a discussion of the presence or absence of excitation of DBOV-Mes at 775 nm (also see comment about page 6 lines 14-15 and Supplementary Figure 9).

We thank the reviewer for the comments. In actual STED imaging for common dyes, both excitation and STED lasers contribute to the bleaching process. However, our study revealed that DBOV-Mes's fluorescence could recover rather than bleach upon exposure to the STED laser alone. Therefore, we initially investigated the impact of the 775 nm STED laser without the excitation laser to determine if the STED beam itself causes photobleaching. As shown in Supplementary Fig. 1, the fluorescence intensity remained largely unchanged before and after continuous exposure to the STED laser, indicating minimal photobleaching effects of the 775 nm STED beam on DBOV-Mes.

We recognize that using only the 775 nm depletion laser without the 561 nm excitation laser may not fully represent the conditions of a long-term STED imaging experiment. To address this, we compared the effects of the 775 nm STED laser in conjunction with excitation (real STED imaging) to excitation alone (confocal imaging), as illustrated in Fig. 3a and b.

In the following response to comment about "*page 6 lines 14-15 and Supplementary Figure 9*", we also provide a more detailed discussion on the possibility that DBOV-Mes may not be excited at 775 nm.

Page 4 lines 3-8, 28-33 and Figure 1B

The explanation about the reactivation of the entire area of ROI1 despite only scanning ROI2 should be moved from lines 28-33 to directly after line 8 to aid with understanding of this result.

We thank the reviewer for this suggestion. We chose to present the experimental observations and provide definitions of "deactivation" and "reactivation" first, followed by an introduction to the possible mechanisms behind these phenomena. The reactivation of the entire area of ROI1 (a larger area), despite only scanning ROI2 (a smaller area), was a side result of the experiment. Therefore, we opted to discuss its explanation separately later in the manuscript to avoid overwhelming the reader with too many details early on. This approach allows readers to grasp the main findings more clearly without being distracted by additional complexity.

Supplementary Figure 4

Define what the red and black arrows show.

We thank the reviewer for pointing this out. We have added one sentence as follows in revised supplementary information:

"Red arrows indicate regions where the absorption decreases, while black arrows highlight areas where the absorption increases after the treatment."

Page 6 lines 14-15 and Supplementary Figure 9

How was it determined that this was an activation resulting from a two-photon effect of the STED beam and not from the STED beam catching the tail end of DBOV-Mes's excitation spectrum? It is earlier stated that the STED laser "does not coincide with the absorption range" (page 2 lines 31-32). Was this confirmed experimentally?

We thank the reviewer for the insightful comments regarding the two-photon effect of the STED beam.

Reply to the second sub-question: As shown in Figure 1A, the normalized absorption spectrum indicates that the DBOV-Mes's absorption peak has a tail, however, there is no absorption observed at 775 nm. This is further corroborated by the raw absorption spectrum presented in Figure R4, which confirms the absence of absorption at 775 nm.

Figure R4. Absorption spectrum of DBOV-Mes (μM in toluene) measured with Perkin-Elmer Lambda 900 spectrometer.

Reply to the first sub-question: We have tested the effect of different STED power densities on DBOV-Mes. The fluorescence retention after different STED power densities scanning on DBOV-Mes was recorded. After prolonged scanning (up to 100 frames, as shown in Supplementary Fig. 11), the fluorescence decreasing was not significant when scanned with 50% STED power (140 mW) compared to 100% STED power (270 mW). This might be induced by the two-photon effect.

We now modified *'two-photon fluorescence was observed when increasing the reactivation beam peak intensity (Supplementary Figs. 10 and 11) ...'* of the revised main text and added the following figure in the revised supporting information:

Supplementary Figure 11: fluorescence retention after different STED power densities scanning

Measured leftover fluorescence ratio over time (STED beam scanned frames). Left: 100% STED power (270 mW measured at objective) was set. Right: 50% STED power (140 mW measured at objective) was set. Experimental details are same as described in Supplementary Figure 1.

Supplementary Figure 11

Clarify if step 2 includes scanning with the 775nm and 561 nm lasers, or 775 nm laser alone.

We thank the reviewer for pointing this out and apologize for the confusion. In supplementary Figure 11, we tested if fluorescence of DBOV-Mes would be affected by long-term scanned with

a high-power pulsed STED beam. To make this clear, we have modified the subtitle of supplementary Figure 11 (13 in revised supplementary) and the described imaging steps as follows:

Supplementary Figure 13: Fluorescence decreases in long-term STED doughnut beam (only 775 nm) scanning

Step 2, images (small, center of image of step 1) were scanned by only STED doughnut beam (775 nm, 270 mW) for 50 frames, 64 pixel X 64 pixel, 0.109 $\mu\text{m}/\text{pixel}$, scan speed 100 Hz, pixel dwell time 102.375 μs , line accumulation of 2.

Page 6 lines 36-39 and Figure 3B

Given the around 100% fluorescence retention in Fig2D when only a few deactivation frames were acquired before reactivation, what explains the rapid drop in fluorescence retention during STED imaging? Would you not expect that deactivation and reactivation would be occurring rapidly within each frame so it should stay at around 100% retention?

We thank the reviewer for the thoughtful comment. In Fig. 2D, we used only 50% STED power to reactivate fluorescence, specifically to demonstrate the repeatability of the deactivation-activation process. During actual STED imaging, however, we employ 100% STED power to fully assess the robustness of DBOV-Mes under more intense conditions. The primary goal of using 100% power is to push the system and highlight DBOV-Mes's stability, rather than to optimize the imaging conditions for fluorescence retention. The rapid drop in fluorescence retention observed during STED imaging could be attributed to the high STED power, which may introduce nonlinear effects such as two-photon fluorescence, leading to faster deactivation.

We acknowledge the reviewer's point and will discuss in the revised manuscript how different STED power settings impact fluorescence retention, and the potential role of nonlinear effects like two-photon fluorescence. We have added such discussion to the revised main text as follows:

“It should be noted that while STED light can enable fluorescence recovery of DBOV-Mes, very high STED power may cause bleaching due to nonlinear effects, making it essential to balance STED power for optimal fluorescence recovery and minimal bleaching.”

Figure 4

It would be good to compare the achievable improvement in resolution to a comparable dye that is optimised for STED e.g. Abberior STAR RED. Likewise, it would also be helpful to show the resolution of the same structure imaged repeatedly with STED to see if there is a degradation of resolution in later images.

We appreciate the reviewer's valuable suggestion.

We have replaced this figure in revised supplementary as follows to show comparison of photostability of DBOV-Mes with abberior STAR RED.

Supplementary Figure 2: Photo-bleaching properties of DBOV by excitation laser

Photo-bleaching properties of DBOV-Mes and STAR RED as a function of the imaging time. Samples were dyes dropped on coverslips and measured in air after the solution evaporated. The measurements were performed with a Leica Stellaris 8 STED microscope. Series of Confocal images were acquired for comparison of photo-bleaching properties. Confocal images (500 frames) were acquired using an excitation wavelength of 610 nm at 20% laser power. Emission was collected across a range from 620 nm to 750 nm. Setting: 64 pixel X 64 pixel, 0.159 $\mu\text{m}/\text{pixel}$, pixel dwell time 102.375 μs , total collection time is 340.059 s.

Page 9 line 17

It does not seem accurate to refer to reconstruction of 2D-STED images into a 3D volume as 3D-STED if there is no improvement in Z resolution. 3D-STED should refer to imaging where there is depletion in both XY plane and the Z-axis.

We thank the reviewer for the insightful comment. We apologize for the confusion regarding the terminology used to describe our imaging approaches.

To clarify, in our study, we have utilized two different types of 3D imaging:

2D-STED with Z-Stack (2D-3D STED): This refers to the reconstruction of a 3D volume from a series of 2D-STED images acquired at different focal planes (Z-stack). Although this method provides 3D information, it does not enhance resolution in the Z-axis beyond the conventional confocal capabilities, as there is no depletion in the Z-axis.

True 3D-STED (3D-3D STED): This involves the use of a 3D-STED beam, which provides depletion in both the XY plane and along the Z-axis, leading to an improvement in resolution in all three dimensions.

To avoid further ambiguity, we revised the manuscript to define these methods as "2D-3D STED" for the former and "3D-3D STED" for the latter. We hope this clarifies the distinction between the two techniques used in our study.

Discussion

The authors refer to examples in the literature showing nanographene biocompatibility etc. which helps to highlight the potential applications of DBOV-Mes. However, they could also include reference to any capabilities to functionalise nanographenes for targeting specific structures of interest.

We thank the reviewer's suggestion. In one of the references (*J. Am. Chem. Soc.* 2024, 146, 5195–5203) we cited, a water-soluble derivative of DBOV-Mes has been successfully functionalized with azide groups, allowing for specific and targeted labeling. However, research on the use of nanographene molecules for fluorescence imaging is still relatively limited, and there are not many published studies in this area yet.

It would also be of significant interest for the authors to discuss the possibility of combining nanographenes with different spectral properties for multi-coloured STED as this would greatly improve the usefulness of these for imaging of biological samples.

We appreciate the reviewer's suggestion to discuss the potential of combining nanographenes with different spectral properties for multi-coloured STED. In particular DBOV-Mes and also other types of synthesized nanographene have very narrow absorption and emission spectra, making them ideal for multi-color imaging.

We have added such discussion to the revised main text as follows:

"The spectra of DBOV derivatives and other types of nanographene obtained through precise synthesis can cover the visible light region, and the narrow absorption and emission spectra make them ideal for multicolor imaging. (J. Am. Chem. Soc. 2019, 141, 41, 16439–16449)"

Note

The photo-chemistry included in this manuscript is outside of my area of expertise and I am therefore unable to comment on this.

Reviewer #4 (Remarks to the Author):

Yang and colleagues report on the use of a red-emitting nanographene, DBOV, in STED microscopy. They discover that DBOV is photostable under STED conditions. Following signal reduction during light exposure, a large portion of fluorescence can be recovered by infrared light exposure. Only with prolonged intense light exposure does a permanent degradation of the signal occur. Hence they term their approach reactivatable STED. “Non-bleaching” or reactivatable fluorophores would be a very valuable addition to the STED toolbox. As such, the study is relevant and timely. The current study is mostly a photophysical characterization of nanographene deactivation and reactivation on a STED microscope. My major point of criticism is that the actual STED imaging performed in the study does not seem to approach the current state of the art performance with organic fluorophores. STED imaging is performed on technical samples (cracks on coverslips, DBOV embedding in polymer) and liposomes, which may point to a difficulty in functionalizing the nanographenes for more general imaging applications. The achieved resolution and – as far as this can be judged from the data provided – signal-to-noise ratio appear lower than with state-of-the-art fluorophores. It should also be discussed what the limitations in resolution and signal to noise ratio are. If photobleaching is not limiting, what determines/limits the quality of the STED images? To make a real difference, ideally, the work would be complemented by high-performance STED measurements in biological samples, including e.g. antibody labeling or live imaging highlighting specific cellular molecules, as this seems to be the promise put forth by the authors. With its current scope, the work may thus be more suited for a more specialized chemistry/biophysics journal.

We thank the reviewer for summarizing our work and for valuable feedback. We would like to address each of the concerns the reviewer raised:

Concern 1. My major point of criticism is that the actual STED imaging performed in the study does not seem to approach the current state of the art performance with organic fluorophores.

We thank the reviewers for this very useful comment. When evaluating the performance of actual STED imaging, several key metrics are usually considered: spatial resolution, signal to noise ratio (SNR), photobleaching, imaging time, laser power, uniformity, depth of field (DOF), spectral crosstalk for multicolor imaging, etc. In this work, we focus on two key metrics, photobleaching and imaging time: 1) our demonstration of nanographene for ReSTED imaging gets rid of the dependence of currently used organic fluorophores on anti-fading agents; and 2) long-term 3D STED imaging time is achieved. From these two points of view, the imaging technology we demonstrated is superior to the state-of-the-art performance of organic fluorophores. In long term studies, we will further investigate other key metrics that go beyond the current work.

Regarding the resolution and signal-to-noise ratio pointed out by the reviewer, we have provided further discussion in our response to “Concern 2” below.

Concern 2. It should also be discussed what the limitations in resolution and signal to noise ratio are. If photobleaching is not limiting, what determines/limits the quality of the STED images?

We thank the reviewer for the insightful comments.

The resolution in STED microscopy is determined by several parameters and can be described by the following formula:

$$d = \frac{\lambda}{2 \cdot NA \cdot \sqrt{1 + \frac{I_{STED}}{I_{sat}}}} \quad (1)$$

Where d is the achievable resolution, λ is the STED beam wavelength, NA is the numerical aperture of the objective lens, I_{STED} is the intensity of the STED beam, I_{sat} is the saturation intensity of the fluorophore, which can be calculated by

$$I_{sat} = \frac{h\nu}{\sigma\tau_{FL}} \quad (2)$$

Where $h\nu$ is the photon energy at the STED wavelength, σ is the cross-section for stimulated emission, τ_{FL} is the fluorescence lifetime of the fluorophore (Nature Photonics 3, 144–147 (2009); Opt. Express 19, 8066-8072 (2011)).

The stimulated emission cross section σ has a spectral dependency,

$$\sigma(\lambda) = \frac{\lambda^4 E(\lambda) \phi}{8\pi c n^2 \tau_{EL}} \quad (3)$$

Where λ is the wavelength, $E(\lambda)$ is the normalized emission spectrum at wavelength λ , ϕ is the fluorescence quantum yield of the fluorophore, c is the speed of light, n is the refractive index, and τ_{EL} is the excited state lifetime (Nanomaterials 2022, 12(1), 21).

From Eq. (1), it can be seen that among all the parameters affecting the imaging resolution, only the saturation intensity I_{sat} of the fluorophore is an intrinsic property of the fluorophore, while the other parameters are determined by the microscope imaging setup. To achieve high resolution, a lower I_{sat} is desirable for the fluorophore. Furthermore, as shown in Eq. (2), obtaining a low I_{sat} requires a long fluorescence lifetime τ_{FL} , and a high stimulated emission cross-section.

In our study of the nanographene DBOV-Mes, the measured fluorescence lifetime is around 3 ns, comparable to that of the other organic dyes. Regarding the stimulated emission cross-section, several factors play a role (as shown in Eq. (3)): the fluorescence quantum yield ϕ , the refractive index n , and the excited state lifetime τ_{EL} , all of which are intrinsic properties of the fluorophore. The typical refractive index n for common organic dyes and carbon dots is around 2 (~1.5-2.4) (Nanoscale, 2022, 14, 8145-8152; RSC Adv., 2017, 7, 36632-36643; Polymers 2021, 13(15), 2545). Notably, Nanographene DBOV-Mes exhibits a very high fluorescence quantum yield of ~80% and a fast excited state lifetime of ~0.5 ps (Mater. Horiz., 2022, 9, 393-402). Consequently, DBOV-Mes is expected to have a high stimulated emission cross-section, which corresponds to a low saturation intensity. Therefore the resolution imaged with nanographene DBOV-Mes should be comparable to the state of the art of organic molecules.

The signal to noise ratio (SNR) quantifies the level of the desired signal in relation to background noise. To achieve a higher signal, it is preferable to use bright fluorophores. As noted in reference

(Angew. Chem. Int. Ed. 2020, 59, 496–502), DBOV-Mes has a brightness of 55300 M⁻¹cm⁻¹, comparable to standard organic dyes like Alexa 647.

Additionally, to maximize fluorescence signal detection, it is crucial to select an appropriate excitation laser wavelength and an emission bandpass filter that aligns with the fluorophore's emission spectrum. Currently, only the commercial Abberior Expert Line microscope available in our laboratory supports true 3D STED imaging. However, this setup utilizes a fixed emission filter of 605-625 nm for red dyes, which unfortunately filters out approximately 68% of the fluorescence signal from our nanographene DBOV-Mes. To enhance the SNR, a tunable emission filter that can be adjusted to match the emission spectrum of DBOV-Mes would be ideal.

As we above described in the response to “Concern 1”, discussing the limitations in resolution and SNR is not the primary focus of the current work. However, to provide relevant information to readers, we have included the following sentences in the revised manuscript:

“Notably, while DBOV-Mes is theoretically capable of achieving high resolution under intense STED laser, it is essential to use an appropriate tunable emission filter for enhancing the signal-to-noise ratio (SNR).”

Concern 3. To make a real difference, ideally, the work would be complemented by high-performance STED measurements in biological samples, including e.g. antibody labeling or live imaging highlighting specific cellular molecules, as this seems to be the promise put forth by the authors.

Text Redacted

Figure Redacted

Minor points

- It would be helpful to exactly describe how individual measurements were performed in figure captions and define all quantities displayed also there.

We thank the reviewer for the suggestion. We understand the importance of clarity in the figure captions; however, adding more text to the captions would make them overly lengthy. To keep the main text concise, we have provided detailed descriptions of all measurements and the definitions of the displayed quantities in the Supplementary.

- Abstract: “tradeoff with imaging time” is slightly confusing. It’s more the number of images that can be acquired.

We thank the reviewer for pointing this out. We now changed the “*tradeoff with imaging time*” to “*tradeoff with total imaging time*”.

- p1, l39: Doughnut shape is not essential to STED, many other light patterns can be used.

We appreciate reviewer’s comment. We now changed the “*doughnut shape*” to “*structured light beams, e.g. commonly used doughnut-shaped*”.

- p2, l6: “high excitation powers”. It’s not just excitation power but also importantly STED laser power that induces photobleaching.

We thank the reviewer for pointing this out, we now change the “*high excitation powers*” with “*high laser powers*”.

- p2, 35: “photon bleaching” should be “photo bleaching”

We thank the reviewer for pointing out this typo, we now replaced “*photon bleaching*” with “*photo bleaching*”.

- p4, l35: Phrasing may be adjusted: Some molecules may also be in the deactivated state at equilibrium. This may potentially be due to ambient light exposure, not just due to the excitation light exposure during focus finding.

We thank the reviewer’s suggestion. We now reworded this to “*This is due to factors such as ambient light which is difficult to avoid and the beam excitation that is required to find the focal plane and select the target imaging area before imaging.*”

- Data in Fig. 5 BD are deconvolved. Rather show original data.

We thank the reviewer’s suggestion.

We presented deconvolved STED images in Figure 5B and 5D to enhance the clarity and resolution of the observed features. Deconvolution is a mathematical algorithm used to correct for the point spread function (PSF) inherent in microscopy, improving the accuracy and sharpness of the image. This is particularly beneficial in distinguishing fine details, which may otherwise be obscured by optical limitations such as blurring and defocusing.

It is important to note that deconvolution corrects for optical aberrations introduced by the microscope and does not alter or artificially enhance the properties of the fluorescent materials themselves. The process is applied purely to improve the visualization of the data, rather than influencing the intrinsic fluorescence characteristics, such as brightness or stability.

However, following the reviewer's suggestion, we are happy to provide the original, non-deconvolved data as Supplementary Figure 24 to allow a direct comparison and ensure full transparency of our results.

Supplementary Figure 24. Non-deconvolved STED images corresponding to Figs. 5 B and D

- Suppl. Fig. 2: It should be stated in the figure caption what imaging buffer was used. Alexa 647 does not seem to be a particularly photostable conventional fluorophore to compare to.

We appreciate the reviewer's valuable suggestion. We have replaced this figure in revised supplementary as follows to show comparison of photostability of DBOV-Mes with abberior STAR RED.

Supplementary Figure 2: Photo-bleaching properties of DBOV by excitation laser

Photo-bleaching properties of DBOV-Mes and STAR RED as a function of the imaging time. Samples were dyes dropped on coverslips and measured in air after the solution evaporated. The measurements were performed with a Leica Stellaris 8 STED microscope. Series of Confocal images were acquired for comparison of photo-bleaching properties. Confocal images (500 frames) were acquired using an excitation wavelength of 610 nm at 20% laser power. Emission was collected across a range from 620 nm to 750 nm. Setting: 64 pixel X 64 pixel, 0.159 $\mu\text{m}/\text{pixel}$, pixel dwell time 102.375 μs , total collection time is 340.059 s.

Reviewer #1 (Remarks to the Author):

After reading the responses to my concerns and new experiments/information/discussion from the authors in the new version of the article, I am satisfied either with the responses to my criticisms as with the changes in the revised version of the manuscript.

I also care of the responses to the other reviewers, and certainly, the work of the authors is mostly convincing to me as well.

In these circumstances, I am in favor of acceptance of the present version of the article in Nat. Comm.

We thank the reviewer for the thoughtful feedback and careful review of our manuscript. We are pleased that the revisions and responses to all concerns, including those of other reviewers, have met the expectations.

Reviewer #2 (Remarks to the Author):

As already stated in my original review of this manuscript, I think it is an excellent piece of work, which has the potential to make a big impact on a burgeoning field in advanced microscopy, namely STED microscopy. In the present, revised version, I find the authors' responses to the reviewers' comments extensive, including new experiments, and predominantly convincing. Importantly, they convinced me that DBOV-OTEG-N3 can be conjugated to a peptide, which enables it to enter the cytoplasm of the root cells of a plant. This success opens the route for the graphene to become a bona fide tool in the hands of cell biologists who need more photostable fluorophores in long-term live cell experiments. Unfortunately, this conjugation also induces a red shift in the graphene's emission, which renders the conjugate non-depletable with the depletion lasers currently available to the authors. For this reason, they resort to showing us confocal images to document their achievement. This is in itself enough for me to accept this additional experiment, although the images only convince me that they got the conjugated graphene into the cells, not really that it labels mitochondria. But that does not take away enough of the overall excellence and impact of the work to request more experiments at this point.

We sincerely thank the reviewer for the high evaluation of our work. We also appreciate the reviewer's understanding regarding the challenges posed by the red shift in the nanographene's emission, which currently limits its compatibility with the depletion lasers available to us. While we were unable to obtain STED images for this reason, we are glad that the confocal images provided were sufficient to demonstrate the successful conjugation and cellular uptake of the nanographene. As the reviewer pointed out, this is a significant step toward the use of graphene-based fluorophores as reliable tools for live-cell imaging.

To address the reviewer's concern about mitochondrial labeling, we recognize that these are preliminary experiments. We are actively collaborating with experts in mitochondrial labeling to

explore this aspect in greater detail. These efforts will be part of our future work, as the focus of the current manuscript lies elsewhere.

Reviewer #3 (Remarks to the Author):

Overall, I am very satisfied with how the authors addressed the questions raised in my initial review. There is one point that was not fully addressed - the question of comparing achievable resolution with a known STED fluorophore such as STAR RED. The authors added in a comparison of bleaching but not of resolution. However, I understand that this would require significant additional work to label the same structure with both dyes so I am happy to recommend that the revised paper be accepted.

We are delighted to receive the reviewer's recognition of our responses.

Regarding the remaining concern about the comparison of resolution with a known STED fluorophore such as STAR RED, as the reviewer mentioned, addressing this would require additional experiments to label the same structure with both dyes. We greatly appreciate the reviewer's suggestion and look forward to addressing this in our future work.

Reviewer #4 (Remarks to the Author):

I appreciate the additional work invested by the authors. I point out that the manuscript is mostly a photophysical study, rather than an actual demonstration of improved performance in practically applicable STED microscopy. Suitability for the journal is ultimately at the editor's discretion.

Below please find specific comments on the revised manuscript.

1) Original comment:

My major point of criticism is that the actual STED imaging performed in the study does not seem to approach the current state of the art performance with organic fluorophores. STED imaging is performed on technical samples (cracks on coverslips, DBOV embedding in polymer) and liposomes, which may point to a difficulty in functionalizing the nanographenes for more general imaging applications.

Authors' response:

We thank the reviewers for this very useful comment. When evaluating the performance of actual STED imaging, several key metrics are usually considered: spatial resolution, signal to noise ratio (SNR), photobleaching, imaging time, laser power, uniformity, depth of field (DOF), spectral crosstalk for multicolor imaging, etc. In this work, we focus on two key metrics, photobleaching and imaging time: 1) our demonstration of nanographene for ReSTED imaging gets rid of the

dependence of currently used organic fluorophores on anti-fading agents; and 2) long-term 3D STED imaging time is achieved. From these two points of view, the imaging technology we demonstrated is superior to the state-of-the-art performance of organic fluorophores. In long term studies, we will further investigate other key metrics that go beyond the current work.

Regarding the resolution and signal-to-noise ratio pointed out by the reviewer, we have provided further discussion in our response to “Concern 2” below.

Comment on response (new comment):

I agree that there are several performance metrics relevant to STED microscopy. I do however point out that these are highly interlinked. E.g. photobleaching strongly depends on the desired resolution. As pointed out above, I thus argue that the present study is an interesting demonstration of photophysics rather than a concrete advance for practical STED imaging.

We thank the reviewer for the insightful feedback. We agree that performance metrics in STED microscopy are highly interconnected. As the reviewer has pointed out, photobleaching is strongly influenced by the desired resolution, with higher resolutions typically requiring higher STED intensities, which can exacerbate photobleaching for commonly used STED imaging dyes.

In this study, our aim is to propose an approach that leverages dyes suitable for STED imaging under prolonged exposure, even at high STED illumination levels. By investigating the mechanisms behind high photostability—specifically, the fluorescence recovery observed under STED laser exposure—we aim to deepen the understanding of the photophysical processes that govern practical imaging.

While the primary focus of our work is on photophysics, we believe that the insights gained have meaningful implications for advancing practical STED imaging by providing guidance for optimizing dye performance and imaging strategies.

2) Original comment:

The achieved resolution and – as far as this can be judged from the data provided – signal-to-noise ratio appear lower than with state-of-the-art fluorophores. It should also be discussed what the limitations in resolution and signal to noise ratio are. If photobleaching is not limiting, what determines/limits the quality of the STED images?

Authors’ response:

We thank the reviewer for the insightful comments.

The resolution in STED microscopy is determined by several parameters and can be described by the following formula:

$$d = \lambda / (2 \cdot NA \cdot \sqrt{1 + I_{\text{STED}} / I_{\text{sat}}}) \quad (1)$$

Where d is the achievable resolution, λ is the STED beam wavelength, NA is the numerical aperture of the objective lens, I_{STED} is the intensity of the STED beam, I_{sat} is the saturation intensity of the fluorophore, which can be calculated by

$$I_{sat} = hv / (\sigma \tau_{FL}) \quad (2)$$

Where hv is the photon energy at the STED wavelength, σ is the cross-section for stimulated emission, τ_{FL} is the fluorescence lifetime of the fluorophore (Nature Photonics 3, 144–147 (2009); Opt. Express 19, 8066-8072 (2011)).

The stimulated emission cross section σ has a spectral dependency,

$$\sigma(\lambda) = (\lambda^4 E(\lambda) \phi) / (8\pi c n^2 \tau_{EL}) \quad (3)$$

Where λ is the wavelength, $E(\lambda)$ is the normalized emission spectrum at wavelength λ , ϕ is the fluorescence quantum yield of the fluorophore, c is the speed of light, n is the refractive index, and τ_{EL} is the excited state lifetime (Nanomaterials 2022, 12(1), 21).

From Eq. (1), it can be seen that among all the parameters affecting the imaging resolution, only the saturation intensity I_{sat} of the fluorophore is an intrinsic property of the fluorophore, while the other parameters are determined by the microscope imaging setup. To achieve high resolution, a lower I_{sat} is desirable for the fluorophore. Furthermore, as shown in Eq. (2), obtaining a low I_{sat} requires a long fluorescence lifetime τ_{FL} , and a high stimulated emission cross-section.

In our study of the nanographene DBOV-Mes, the measured fluorescence lifetime is around 3 ns, comparable to that of the other organic dyes. Regarding the stimulated emission cross-section, several factors play a role (as shown in Eq. (3)): the fluorescence quantum yield ϕ , the refractive index n , and the excited state lifetime τ_{EL} , all of which are intrinsic properties of the fluorophore. The typical refractive index n for common organic dyes and carbon dots is around 2 (~1.5-2.4) (Nanoscale, 2022, 14, 8145-8152; RSC Adv., 2017, 7, 36632-36643; Polymers 2021, 13(15), 2545). Notably, Nanographene DBOV-Mes exhibits a very high fluorescence quantum yield of ~80% and a fast excited state lifetime of ~0.5 ps (Mater. Horiz., 2022, 9, 393-402). Consequently, DBOV-Mes is expected to have a high stimulated emission cross-section, which corresponds to a low saturation intensity. Therefore the resolution imaged with nanographene DBOV-Mes should be comparable to the state of the art of organic molecules.

Comment on response (new comment):

The authors recapitulate the well-known square root dependence of STED resolution on intensity, with the saturation intensity setting the scale. It would be informative and substantially strengthen the manuscript to actually measure the saturation intensity and/or give resolution measurements that are on a par with the resolution achieved with organic fluorophores.

We thank the reviewer for the insightful comment. The saturation intensity in STED microscopy is typically determined by gradually increasing the STED laser power and monitoring the change

in fluorescence intensity to identify the optimal STED power. However, in our ReSTED approach, the STED light does not only contribute to fluorescence depletion; it also enables fluorescence recovery. As a result, the behavior of the dye in our method differs from that of commonly used dyes. For typical dyes, the fluorescence intensity decreases with increasing STED power, eventually reaching a saturation point where further increases in STED intensity result in only minimal changes in fluorescence. In contrast, the fluorescence recovery in our method does not follow this conventional pattern, which makes it unsuitable to use the same approach for determining the saturation power in STED microscopy.

As such, in our previous response, we used known formulas to estimate that the resolution achieved with nanographene DBOV-Mes should be comparable to that of state-of-the-art organic fluorophores. Furthermore, we agree that a direct comparison of resolution with a well-established STED fluorophore, such as STAR RED, would be informative. However, addressing this would require additional experiments, such as labeling the same structure with both dyes. We greatly appreciate the reviewer's suggestion and look forward to pursuing this comparison in our future work.

Authors' response:

The signal to noise ratio (SNR) quantifies the level of the desired signal in relation to background noise. To achieve a higher signal, it is preferable to use bright fluorophores. As noted in reference (Angew. Chem. Int. Ed. 2020, 59, 496–502), DBOV-Mes has a brightness of 55300 M⁻¹cm⁻¹, comparable to standard organic dyes like Alexa 647.

Additionally, to maximize fluorescence signal detection, it is crucial to select an appropriate excitation laser wavelength and an emission bandpass filter that aligns with the fluorophore's emission spectrum. Currently, only the commercial Abberior Expert Line microscope available in our laboratory supports true 3D STED imaging. However, this setup utilizes a fixed emission filter of 605-625 nm for red dyes, which unfortunately filters out approximately 68% of the fluorescence signal from our nanographene DBOV-Mes. To enhance the SNR, a tunable emission filter that can be adjusted to match the emission spectrum of DBOV-Mes would be ideal.

As we above described in the response to "Concern 1", discussing the limitations in resolution and SNR is not the primary focus of the current work. However, to provide relevant information to readers, we have included the following sentences in the revised manuscript:

"Notably, while DBOV-Mes is theoretically capable of achieving high resolution under intense STED laser, it is essential to use an appropriate tunable emission filter for enhancing the signal-to-noise ratio (SNR)."

Comment on response (new comment):

For collecting more of the emitted fluorescence, it is not necessary to have a tunable filter. Merely exchanging the detection bandpass filter would be sufficient, which is easily possible in the Abberior Expert Line microscope. The authors point to a limitation for 3D STED imaging. 3D STED imaging is not necessary for measuring lateral resolution, which would be sufficient in this case.

We sincerely appreciate the reviewer's valuable suggestion. Although it is feasible to change the detection bandpass filter on the Abberior Expert Line microscope, we chose not to do so in current work in order to maintain consistency of our all experiments. On the other side, this was intended as a validation experiment, demonstrating that the dye can be used for long-term, high-power STED imaging. Regarding the achievable resolution, we greatly appreciate the reviewer's suggestion and look forward to addressing this in our future work.

3) Original Comment:

To make a real difference, ideally, the work would be complemented by high-performance STED measurements in biological samples, including e.g. antibody labeling or live imaging highlighting specific cellular molecules, as this seems to be the promise put forth by the authors.

Comment:

The authors indicate that STED measurements are not possible with current biocompatible variants due to a spectral shift. I appreciate this limitation.

We thank the reviewer for understanding regarding the limitation of STED measurements with the current biocompatible variants due to the spectral shift. We appreciate the reviewer's acknowledgment of this challenge. We are hopeful that future advancements, either in the development of biocompatible fluorophores or in the design of home-built microscopes with suitable wavelength ranges, will help overcome this issue and enable STED measurements with these materials.

Further comments:

Supplementary Fig. 1:

„The fluorescence intensity after scanned by the STED beam was even higher than in the first image because additional beam excitation is required to find the focal plane and select the target imaging area before imaging. Therefore, before taking the first confocal image, some DBOV-Mes molecules may already be in the fluorescent OFF state.“

I deem it equally likely that there is a fraction of molecules in a non-fluorescent state in equilibrium. This population may get activated by the STED beam, resulting in higher intensity after scanning with the STED beam.

We agree with the reviewer that some molecules may also be in the deactivated state at equilibrium. This may potentially be due to ambient light exposure, not just due to the excitation light exposure during focus finding.

We now reworded this to *“The fluorescence intensity after scanned by the STED beam was even higher than in the first image. This is due to factors such as ambient light which is difficult to avoid and the beam excitation that is required to find the focal plane and select the target imaging area before imaging. Therefore, before taking the first confocal image, some DBOV-Mes molecules may already be in the fluorescent OFF state.”*

Supplementary Fig. 2:

Are photobleaching properties of Abberior STAR RED after evaporation of the solution similar as in the more usual setting of aqueous solution?

We thank the reviewer for the insightful question.

Our primary focus for this measurement was to showcase the photostability of nanographene. To ensure a direct and unbiased comparison, both nanographene and Abberior STAR RED were evaluated under identical conditions, without the use of an antifading solution.

The "more usual setting of aqueous solution" typically includes the use of antifading agents for Abberior STAR RED imaging, which are known to significantly improve its photobleaching resistance, particularly under high laser power. Without antifading agents, the photobleaching performance of Abberior STAR RED in aqueous solutions would likely be significantly worse—similar to what we observed in air. Unfortunately, due to limitations in immobilizing Abberior STAR RED on substrates in aqueous environments (both with and without antifading solutions) using our current sample preparation methods, such measurements were not feasible as they would induce diffusion of the dyes.